# PathWise: Planning through World Model for Automated Heuristic Design via Self-Evolving LLMs

**Oguzhan Gungordu**[1]   **Siheng Xiong**[1]   **Faramarz Fekri**[1]

## Abstract

Large Language Models (LLMs) have enabled automated heuristic design (AHD) for combinatorial optimization problems (COPs), but existing frameworks' reliance on fixed evolutionary rules and static prompt templates often leads to myopic heuristic generation, redundant evaluations, and limited reasoning about how new heuristics should be derived. We propose a novel multi-agent reasoning framework, referred to as **Pla**nning **th**rough **W**orld Model for Automated Heuris**t**ic Design via **S**elf-**E**volving LLMs (Path-Wise), which formulates heuristic generation as a sequential decision process over an *entailment graph* serving as a compact, stateful memory of the search trajectory. This approach allows the system to carry forward past decisions and reuse or avoid derivation information across generations. A policy agent plans evolutionary actions, a world model agent generates heuristic rollouts conditioned on those actions, and critic agents provide routed reflections summarizing lessons from prior steps, shifting LLM-based AHD from trial-and-error evolution toward state-aware planning through reasoning. Experiments across diverse COPs show that PathWise converges faster to better heuristics, generalizes across different LLM backbones, and scales to larger problem sizes.

## 1. Introduction

Heuristic algorithms are central to solving COPs, which arise in many real-world complex decision-making tasks such as routing, scheduling, logistics, and design automation (Desale et al., 2015; Ma et al., 2019; Tam et al., 2024). Since many COPs are NP-hard, heuristics are often the only practical approach for obtaining high-quality solutions

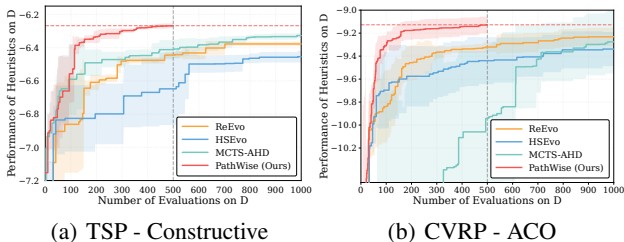

(a) TSP - Constructive         (b) CVRP - ACO

*Figure 1.* Evolution curves of LLM-based AHD methods on (a) TSP and (b) CVRP under different search frameworks, showing best-so-far heuristic performance as a function of evaluation number for representative population-based and tree-based methods using `GPT-4o-mini`. PathWise is run with a limit of $n_e = 500$ evaluations while all baselines use $n_e = 1000$, yet achieves stronger performance with lower variance and faster convergence.

within reasonable time, leading to the widespread adoption of methods such as simulated annealing, tabu search, and iterated local search (Kirkpatrick et al., 1983; Glover, 1990; Lourenço et al., 2003). Despite their success, constructing effective heuristics remains manual, requiring substantial domain expertise to design solver-specific algorithmic components, making the process costly and difficult to generalize across problems (Choong et al., 2018; Pillay & Qu, 2018). To address these challenges, AHD has emerged as a framework for automatically generating heuristics for a given problem class, aiming to reduce dependence on expert-crafted designs and enabling systematic discovery through Genetic Programming (GP) (Burke et al., 2013; Langdon & Poli, 2013). However, GP-based AHD methods represent heuristics as syntax trees and rely on evolutionary operators confined to human-defined arithmetic operators, limiting their flexibility and effectiveness (Duflo et al., 2019).

Recently, LLMs have shown capability in reasoning and code generation, opening new directions for AHD (Chen et al., 2021; Gandhi et al., 2023; Yang et al., 2024b). Given these capabilities, LLM-based AHD methods integrate LLMs into evolutionary search to automatically generate heuristics with minimal human intervention (Liu et al., 2023). Many approaches adopt population-based evolutionary procedures, where LLMs iteratively refine a population of candidate algorithms using evolutionary

---

[1]Georgia Institute of Technology. Correspondence to: Oguzhan Gungordu <ogungordu3@gatech.edu>.

*Proceedings of the 43rd International Conference on Machine Learning*, Seoul, South Korea. PMLR 306, 2026. Copyright 2026 by the author(s).

search (Chen et al., 2023; Meyerson et al., 2024). Early methods such as FunSearch (Romera-Paredes et al., 2023) and EoH (Liu et al., 2024a) employ LLM-guided operators to evolve high-performing heuristics, enabling fast heuristic search. ReEvo (Ye et al., 2024) further incorporates a reflection mechanism to analyze generated heuristics and guide search (Shinn et al., 2023). Following this, HSEvo (Dat et al., 2025) introduces diversity-aware population management and harmony search to improve population diversity (Shi et al., 2013). Beyond population-based approaches, tree-based methods have been explored (Wang et al., 2025a). Specifically, MCTS-AHD (Zheng et al., 2025) integrates LLMs with Monte Carlo Tree Search (MCTS), applying the UCT algorithm to guide selection and expansion of heuristic nodes in a tree structure to search the heuristic space (Świechowski et al., 2023).

However, existing LLM-based AHD methods are limited by how heuristic search is structured. Population-based approaches rely on fixed selection and replacement rules, which can lead to premature convergence by discarding intermediate heuristics. Tree-based methods, such as MCTS-AHD, impose a hierarchical structure, but selection and expansion are driven by performance-based UCT criteria instead of semantic understanding of the search landscape or the relationships between heuristics. Exploration is guided by visitation statistics instead of distinctions between heuristic transformations, and node expansion follows a single-path, with long training time. Overall, both population-based and tree-based frameworks treat heuristic generation as isolated or statistically linked sampling steps and rely on fixed sets of operators or prompt templates that do not adapt to evolving search dynamics or different problem settings (van Stein et al., 2025). They lack a stateful, semantic representation of how heuristics are derived, how edits propagate across generations, or why modifications succeed or fail. This lack of memory and state-aware planning leads to redundant evaluations, similar heuristic rediscovery, and inefficient use of LLM calls (Liu et al., 2025d) (see Figure 1).

To address these limitations, we propose **Pl**anning **th**rough **W**orld Model for Automated Heuristic Design via **S**elf-**E**volving LLMs (PathWise), a *structured multi-agent reasoning framework* formulating heuristic discovery as a sequential decision process over an *entailment graph* (Dalvi et al., 2021; Xiong et al., 2025b). The entailment graph provides a compact, stateful representation of the search trajectory, capturing how heuristics are derived and how edits compose across generations, enabling memory and state-aware planning to guide heuristic evolution.

We summarize our major contributions as follows. **(1)** We introduce a hybrid graph-based and population-based formulation of heuristic evolution, where an entailment graph encodes derivation rationale, parent information, and perfor-mance history, serving as a shared state through which the policy and world model interact to guide heuristic discovery. **(2)** We propose a coordinated multi-agent LLM framework where a policy agent controls high-level evolutionary strategy by generating evolutionary actions, including parent selection and derivation rationale, a world model executes these actions through low-level heuristic generation to update the entailment graph, and critic agents analyze graph structure and provide routed reflections that adapt the behavior of the policy and world model, enabling self-evolving, state-aware heuristic generation. **(3)** To ensure variety during heuristic generation, we introduce prompt-level diversity at the level of policy actions and world model rollouts, enabling broader exploration over the entailment graph. **(4)** Through extensive experiments across diverse COPs, we demonstrate that PathWise consistently discovers stronger heuristics using fewer evaluations, achieves faster convergence, and scales more effectively to larger problem sizes.

## 2. Preliminaries

### 2.1. LLM-based AHD for Combinatorial Optimization

AHD considers a COP with instance space $\mathcal{X}$ and solution space $\mathcal{S}$. A heuristic is an executable program $h \in \mathcal{H}$ that maps each instance to a feasible solution, $h : \mathcal{X} \to \mathcal{S}$, where $x \in \mathcal{X}$ denotes a problem instance and $s = h(x) \in \mathcal{S}$ is the corresponding solution. A cost function $f : \mathcal{S} \to \mathbb{R}$ evaluates solution quality. For example, in the Traveling Salesman Problem, an instance provides distance matrix, the solution is a tour, and the cost is its total length.

Performance is assessed over a distribution or dataset $\mathcal{D}$ of instances using the expected negative cost

$$P(h; \mathcal{D}) = \mathbb{E}_{x \sim \mathcal{D}}\big[ - f(h(x))\big], \tag{1}$$

so higher values of $P(h; \mathcal{D})$ correspond to stronger heuristics. Under an evaluation budget, AHD aims to identify

$$h^\star = \arg\max_{h \in \mathcal{H}} P(h; \mathcal{D}). \tag{2}$$

While the space $\mathcal{H}$ encompasses diverse heuristics, searching for a full solver implementation from scratch is often inefficient. Instead, LLM-based AHD methods design a heuristic function $h$ within a chosen search framework.

### 2.2. Structured Reasoning via Entailment Graphs.

We conceptualize heuristic discovery as multi-step reasoning for solving a search problem via the construction of an entailment graph $\mathcal{G} = (\mathcal{V}, \mathcal{E})$. Each node $v \in \mathcal{V}$ represents a tuple $(h, \kappa, d, P(h; \mathcal{D}), \mathrm{PM})$, consisting of heuristic code $h$, a natural-language derivation rationale $\kappa$ used to generate $h$, a natural-language algorithmic description $d$, performance $P(h; \mathcal{D})$, and compact parent metadata $\mathrm{PM}$. Each directed

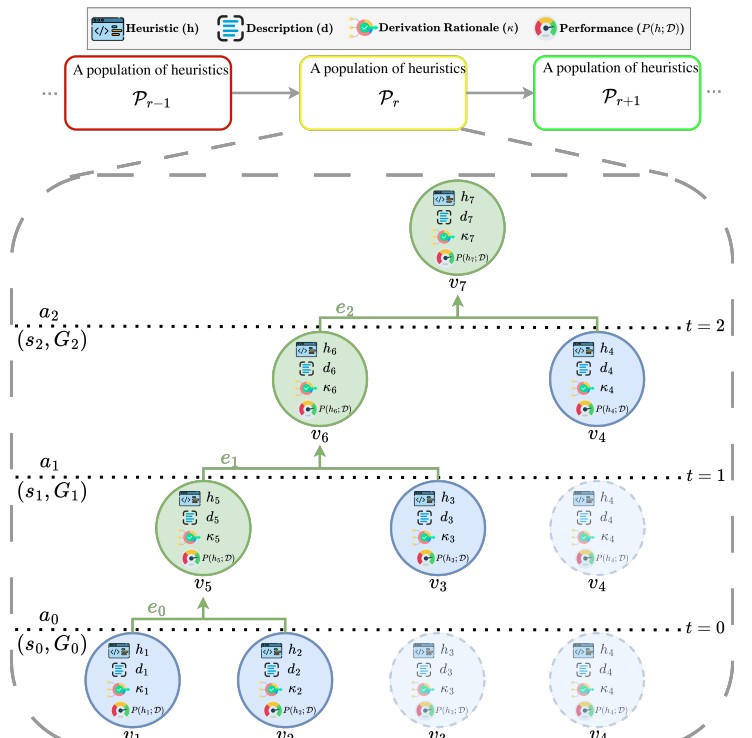

(a) Population-based search combined with entailment-graph reasoning at outer iteration $r$. The graph $G_t = (V_t, E_t)$ expands over inner steps $t$ starting from population $\mathcal{P}_r$, where new nodes are derived from selected parents and added to the graph. The next population $\mathcal{P}_{r+1}$ is formed from nodes generated in $r$.

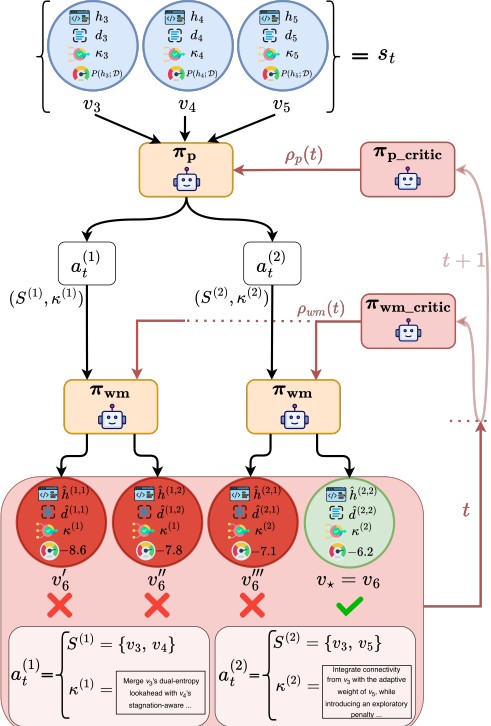

(b) Illustration of an entailment step. The policy agent proposes actions, the world model generates rollouts, and critic agents analyze the resulting outputs at step $t$ to produce reflections conditioning agents at step $t+1$.

*Figure 2.* Overview of *PathWise*. (a) AHD is orchestrated across two timescales, with inner entailment steps within each outer iteration; example shown with $N_p = 4$ (parent metadata in nodes omitted for simplicity). (b) Entailment step showing policy and world model interaction with critics, carrying forward lessons to guide heuristic generation; illustrated at $t = 1$ resulting in the entailment of $v_6$.

edge $e \in \mathcal{E}$ connects a parent set $S$ to the child node $v$, encoding *how* the heuristic was derived from its parents. The entailment graph enables conditioning future decisions on derivation history rather than independent steps, providing a structured memory of the search process.

### 2.3. Sequential Decision View of Heuristic Evolution

We formulate the heuristic discovery as a sequential Markov Decision Process (MDP) rather than a stateless evolutionary algorithm. The search iteratively constructs an *entailment graph* and is represented as $(\mathbb{S}, \mathbb{A}, \mathbb{T}, \mathbb{R})$, where:

- **State $\mathbb{S}$.** The state $s_t \in \mathbb{S}$ corresponds to the current *entailment graph* with its active frontier of nodes available for selection (Figure 2(a)). This structured state encodes how existing heuristics were derived, providing a compact, stateful memory of the search trajectory without requiring access to the entire search history.

- **Action $\mathbb{A}$.** An action $a_t \in \mathbb{A}$ is defined as a tuple $(S, \kappa)$, where $S \subseteq s_t$ is a set of nodes selected from

the current state $s_t$, and $\kappa$ is a natural-language *derivation rationale* generated by the policy (Figure 2(b)). Rather than selecting from a fixed set of rigid operators, $\kappa$ specifies how the selected heuristics should be transformed or combined to generate a child heuristic.

- **Transition $\mathbb{T}$.** The transition $\mathbb{T}(s_t, a_t)$ corresponds to an entailment step (Figure 2(a)) resulting in $s_{t+1}$. Given $(S, \kappa)$, the world model generates a new heuristic $h$ and its algorithmic description $d$, inserts a corresponding node into the entailment graph, and adds an edge recording derivation from $S$ under directive $\kappa$.

- **Reward $\mathbb{R}$.** The reward function $\mathbb{R}(s_t, a_t)$ measures the quality of action $a_t$ given state $s_t$ and generated heuristics (Figure 2(b)). It serves as a signal for evolving agents, which translate performance values into natural-language feedback for the next step.

PathWise is training-free in the sense that no LLM parameters are updated. The MDP formulation serves as a structural backbone for state-aware planning, where the policy and

world model are reinforced through natural-language reflections from critic agents rather than gradient-based policy optimization. Following (Hao et al., 2023), we use the term *world model* to denote an LLM that predicts the next state given the current state and action through its pretrained knowledge, distinct from learned-dynamics models.

## 3. Methodology

PathWise orchestrates heuristic discovery across two timescales using a *hybrid graph-based and population-based* approach (Figure 2(a)). The *outer loop*, indexed by $r$, maintains a population $\mathcal{P}_r$ of root nodes of size $\mathcal{N}_p$. Initial population $\mathcal{P}_0$ is formed by prompting an LLM with an initialization prompt to generate $\mathcal{N}_p$ candidates. Within each outer iteration, an *inner loop*, indexed by $t$, explores the local search space by incrementally constructing an entailment graph $G_t$ rooted at $\mathcal{P}_r$. At each inner step, the entailment graph is expanded through an entailment operation that derives a new node from its selected parent nodes while recording the corresponding derivation relationship. Each entailment operation is realized by coordinated LLM agents that jointly perform planning, code synthesis, and reflective feedback (Figure 2(b)), with their roles and interaction detailed in the remainder of this section.

Unlike population-based evolutionary methods that discard intermediate results, derivation history is retained in the graph structure, compressing the search trajectory. Additionally, unlike tree-based approaches that preserve all generated heuristics and require selecting among ever-growing candidate sets, the population mechanism restricts exploration to a compact set of root nodes at each outer iteration.

### 3.1. Entailment Graph Construction

**State Representation.** We denote the entailment graph at inner step $t$ as $G_t = (V_t, E_t)$. At the start of each inner loop, the graph is initialized with the current population of root nodes, setting $s_0 = V_0 = \mathcal{P}_r$ and $E_0 = \emptyset$. As entailment proceeds, each newly generated node $v$ at step $t$, derived from a parent set $S \subseteq s_t$, is represented by a tuple $(h, \kappa, d, P(h; \mathcal{D}), \mathrm{PM})$. The parent metadata $\mathrm{PM}$ provides a compressed summary of the parent set $S$, and is defined as $\mathrm{PM} = \{(d_k, P(h_k; \mathcal{D})) \mid v_k \in S\}$, recording compact algorithmic descriptions and performance values of the parent heuristics. This compressed representation allows the model to condition on how a heuristic was derived and on the relative performance of its parents, without including full parent code in the context, thereby minimizing context usage when prompting LLMs.

**Entailment and State Transition.** The entailment graph is incrementally constructed through a sequence of entailment operations. At each inner step $t$, the system entails a new

node $v_\star$ and transitions from $G_t$ to $G_{t+1}$. This operation is represented by a directed edge $S \overset{\kappa}{\Rightarrow} v_\star$, indicating that the parent set $S \subseteq s_t$ entails $v_\star$ under derivation rationale $\kappa$. To control search complexity, the state is updated after each entailment step according to $s_{t+1} = (s_t \cup \{v_\star\}) \setminus (S^{(i_\star)} \setminus \{v^\star\})$, where the newly entailed node $v_\star$ is added, the parent nodes used in the selected entailment are removed, and the global best node $v^\star$ is retained. This update rule balances exploration via entailment, exploitation via the retained global best, and complexity control via pruning of used parent nodes, keeping the state compact.

### 3.2. Multi-Agent Entailment Step.

PathWise employs coordinated LLM agents for heuristic discovery, interacting through a cycle of planning, execution, and reflection to navigate the entailment graph construction. A *Policy Agent* $\boldsymbol{\pi}_{\mathbf{p}}$ observes state $s_t$ and proposes actions by selecting a parent set $S$ and formulating a derivation rationale $\kappa$. A *World Model Agent* $\boldsymbol{\pi}_{\mathbf{wm}}$ executes each action by generating heuristic rollouts. Two critic agents guide this process: a *Policy Critic* $\boldsymbol{\pi}_{\mathbf{p\_critic}}$ reflects on evolutionary strategy, and a *World Model Critic* $\boldsymbol{\pi}_{\mathbf{wm\_critic}}$ reflects on heuristic generation quality. Their feedback guides state-aware planning over the entailment graph, as illustrated in Figure 2(b). We sample $N_a$ actions per inner step and generate $N_w$ heuristic rollouts per action to insert a single entailed node $v_\star$ into the entailment graph. The prompt templates used by agents are provided in Appendix F.1

**Policy Agent Sampling.** The Policy Agent $\boldsymbol{\pi}_{\mathbf{p}}$ acts as the high-level planner. Conditioned on the current state $s_t$ and the routed policy reflection $\rho_p(t)$, it samples $N_a$ candidate entailment actions $\{a_t^{(i)}\}_{i=1}^{N_a}$ according to

$$ a_t^{(i)} = (S^{(i)}, \kappa^{(i)}) \sim \boldsymbol{\pi}_{\mathbf{p}}\big(\cdot \mid s_t, \rho_p(t)\big). \tag{3} $$

Each action $a_t^{(i)} = (S^{(i)}, \kappa^{(i)})$ consists of a parent set $S^{(i)} \subseteq s_t$ and a natural-language directive $\kappa^{(i)}$ that specifies how the selected heuristics should be modified or combined. For each action, the parent metadata is defined from the selected parent nodes as $\mathrm{PM}^{(i)} = \{(d_k, P(h_k; \mathcal{D})) \mid v_k \in S^{(i)}\}$. This action design enables the policy to operate at a semantic level, reasoning over evolutionary strategy. Consequently, the policy can dynamically invent new operator types through $\kappa^{(i)}$ and adapt its edit strategy to the evolving search landscape, instead of using fixed operator templates.

**World Model Rollouts.** The World Model Agent $\boldsymbol{\pi}_{\mathbf{wm}}$ acts as a low-level executor that translates the policy's high-level actions into code-level heuristics. For each policy-generated action $a_t^{(i)}$, it generates $N_w$ candidate rollouts $\{(\hat{h}^{(i,j)}, \hat{d}^{(i,j)})\}_{j=1}^{N_w}$ conditioned on the selected parents, the

directive, and the routed world model reflection $\rho_{wm}(t)$:

$$(\hat{h}^{(i,j)}, \hat{d}^{(i,j)}) \sim \boldsymbol{\pi}_{\mathbf{wm}}\Big(\cdot \mid \{(h_k, d_k)\}_{v_k \in S^{(i)}}, \kappa^{(i)}, \rho_{wm}(t)\Big).$$
(4)

Each heuristic rollout is evaluated on $\mathcal{D}$, and the best-performing candidate $(i_\star, j_\star) = \arg\max_{i,j} P(\hat{h}^{(i,j)}; \mathcal{D})$ is selected. We denote the corresponding heuristic and its algorithmic description as $h_\star = \hat{h}^{(i_\star, j_\star)}$ and $d_\star = \hat{d}^{(i_\star, j_\star)}$. The heuristic is inserted into the entailment graph as an entailed node $v_\star = (h_\star, \kappa^{(i_\star)}, d_\star, P(h_\star; \mathcal{D}), \mathrm{PM}^{(i_\star)})$ and added via the edge $S^{(i_\star)} \xrightarrow{\kappa^{(i_\star)}} v_\star$, yielding $V_{t+1} \leftarrow V_t \cup \{v_\star\}$ and $E_{t+1} \leftarrow E_t \cup \{S^{(i_\star)} \xrightarrow{\kappa^{(i_\star)}} v_\star\}$. It extends the graph and conditions subsequent policy and world model decisions.

### 3.3. Diversity in Policy and World Model Rollouts

A key bottleneck in PathWise occurs when, at an inner step $t$, the policy and world model produce low-diversity outputs, resulting in similar parent selections, directives, and heuristic rollouts. When the policy repeatedly selects the same parent sets with similar directives and the world model generates nondiverse rollouts, the critic agents observe little contrast and cannot generate informative reflections. To address this, we introduce two diversity mechanisms that operate at the prompt level and deliberately alter the posterior distributions of $\boldsymbol{\pi}_{\mathbf{p}}$ and $\boldsymbol{\pi}_{\mathbf{wm}}$ without modifying the underlying graph topology.

**Diversity-Aware Prompt Perturbation.** To encourage diverse sampling from both agents, we introduce a prompt-level perturbation mechanism controlled by a time-varying exploration rate $\varepsilon(\ell)$. Each agent is associated with a role-specific inventory of exploratory phrases, $\Phi_p$ and $\Phi_{wm}$, designed to diversify action proposals and heuristic rollout generation. The exploration rate decays linearly with evaluation count $\ell \in \{0, \ldots, n_e\}$, with $\varepsilon^{\mathrm{init}} = 0.5$ and $\varepsilon^{\mathrm{final}} = 0.25$:

$$\varepsilon(\ell) = \varepsilon^{\mathrm{init}} + (\varepsilon^{\mathrm{final}} - \varepsilon^{\mathrm{init}}) \cdot \ell/n_e.$$
(5)

During sampling, the prompts for $\boldsymbol{\pi}_{\mathbf{p}}$ and $\boldsymbol{\pi}_{\mathbf{wm}}$ are perturbed by sampling $\phi^{(i)} \sim \Phi_p$ and $\psi^{(i,j)} \sim \Phi_{wm}$ with probability $\varepsilon(\ell)$, respectively. These perturbations reshape the sampling distributions, promoting broader exploration early in search and more focused refinement later. The full phrase inventories are provided in Appendix F.2.

**State Shuffling.** To reduce positional bias, where LLMs tend to select parents appearing earlier in the context window (Li et al., 2024; Schilcher et al., 2025; Bito et al., 2025), we randomly permute the order of nodes in $s_t$ before prompting $\boldsymbol{\pi}_{\mathbf{p}}$ and $\boldsymbol{\pi}_{\mathbf{p\_critic}}$, encouraging parent selection to be driven by semantic suitability rather than positional effects.

### 3.4. Critic Models and Routed Reflections

We employ two critic agents, $\boldsymbol{\pi}_{\mathbf{p\_critic}}$ and $\boldsymbol{\pi}_{\mathbf{wm\_critic}}$, to synthesize verbal gradients (Shinn et al., 2023; Madaan

et al., 2023) from the outputs generated by the agents at step $t$. Based on these outputs, the critics generate routed reflections that condition the policy and world model at step $t+1$, enabling state-aware adaptation over the entailment graph without updating LLM parameters.

**Policy Critic ($\boldsymbol{\pi}_{\mathbf{p\_critic}}$).** The policy critic reflects on the evolutionary strategy induced by the actions generated by the policy. For each action $a_t^{(i)}$, it aggregates the performance values of the associated heuristic rollouts into

$$R_p(a_t^{(i)}) = \frac{1}{N_w} \sum_{j=1}^{N_w} P(\hat{h}^{(i,j)}; \mathcal{D}),$$

which summarizes action effectiveness and serves as a reward signal to *rank* actions. Together with a per-action rollout descriptor bundle $\mathcal{B}_p^{(i)} = \{(\hat{d}^{(i,j)}, P(\hat{h}^{(i,j)}; \mathcal{D}))\}_{j=1}^{N_w}$, which compactly captures variation across rollouts, the critic produces a policy reflection

$$\rho_p(t+1) = \boldsymbol{\pi}_{\mathbf{p\_critic}}\Big(\{(a_t^{(i)}, R_p(a_t^{(i)}), \mathcal{B}_p^{(i)})\}_{i=1}^{N_a}, s_t\Big).$$
(6)

By analyzing how different derivation rationales and parent selections interact with the current topology and performance landscape, the policy critic injects routed feedback to $\boldsymbol{\pi}_{\mathbf{p}}$ to adjust its strategy at step $t+1$.

**World Model Critic ($\boldsymbol{\pi}_{\mathbf{wm\_critic}}$).** The world model critic reflects on code-synthesis quality by contrasting the best-performing heuristic rollout with the worst-performing one. The index of the worst-performing rollout is identified as $(i_{\min}, j_{\min}) = \arg\min_{i,j} P(\hat{h}^{(i,j)}; \mathcal{D})$. Using the best-performing rollout $(i_\star, j_\star)$ and the corresponding worst-performing rollout, we form the comparison tuples $\mathbf{best} = (\hat{h}^{(i_\star, j_\star)}, \hat{d}^{(i_\star, j_\star)}, P(\hat{h}^{(i_\star, j_\star)}; \mathcal{D}))$ and $\mathbf{worst} = (\hat{h}^{(i_{\min}, j_{\min})}, \hat{d}^{(i_{\min}, j_{\min})}, P(\hat{h}^{(i_{\min}, j_{\min})}; \mathcal{D}))$. Using these inputs, the critic produces a routed reflection

$$\rho_{wm}(t+1) = \boldsymbol{\pi}_{\mathbf{wm\_critic}}(\mathbf{best}, \mathbf{worst}).$$
(7)

This world model reflection $\rho_{wm}$ conditions $\boldsymbol{\pi}_{\mathbf{wm}}$ at step $t+1$, guiding its code-synthesis through heuristic contrast.

### 3.5. Evolutionary Cycle and Population Management

The inner entailment graph expansion runs until either the maximum number of entailment steps $I_{\max}$ is reached or no further entailment actions are possible (i.e., the frontier $s_t$ collapses to a single node). Upon completion of the inner loop at its final step $t' \in \{1, \ldots, I_{\max} - 1\}$, the resulting entailment graph $G_{t'}$ captures the trajectory explored during the inner loop. The population $\mathcal{P}_{r+1}$ for the next outer iteration is then formed using a *leaf-first selection strategy*.

Let $\mathcal{F} = \mathrm{LeafNodes}(G_{t'})$ denote the set of leaf nodes with no outgoing edges that involved in at least one entailment

*Table 1.* Comparison of methods designing step-by-step construction heuristics on TSP and KP (7 test sets, 250 instances each). LKH-3 (Helsgaun, 2017) and OR-Tools (Perron & Furnon, 2025) provide optimal baselines for TSP and KP, respectively. We report mean performance over 3 runs for each LLM-based AHD method. The best-performing method for each LLM is shaded, and each test set's overall best result is shown in bold. Entries marked "N/A" indicate that the method was not available for that task.

| Task | TSP | | | | | | KP | | | | | | | |
|---|---|---|---|---|---|---|---|---|---|---|---|---|---|---|
| Test sets | $N{=}50$ | | $N{=}100$ | | $N{=}200$ | | $N{=}50, W{=}12.5$ | | $N{=}100, W{=}25$ | | $N{=}200, W{=}25$ | | $N{=}500, W{=}25$ | |
| Methods | Obj.↓ | Gap | Obj.↓ | Gap | Obj.↓ | Gap | Obj.↑ | Gap | Obj.↑ | Gap | Obj.↑ | Gap | Obj.↑ | Gap |
| Optimal | 5.687 | - | 7.767 | - | 10.709 | - | 20.089 | - | 40.254 | - | 57.132 | - | 90.763 | - |
| Greedy Construct | 6.992 | 22.95% | 9.702 | 24.91% | 13.430 | 25.41% | 20.033 | 0.28% | 40.205 | 0.12% | 57.079 | 0.09% | 90.712 | 0.06% |
| POMO | **5.711** | **0.42%** | **8.028** | **3.36%** | 13.014 | 21.52% | 19.669 | 2.09% | 39.626 | 1.56% | 56.909 | 0.39% | 87.931 | 3.12% |
| LLM-based AHD: *GPT-4o-mini* | | | | | | | | | | | | | | |
| Funsearch | 6.452 | 13.45% | 9.050 | 16.52% | 12.806 | 19.58% | 20.037 | 0.26% | 40.190 | 0.16% | 57.035 | 0.17% | 90.110 | 0.72% |
| EoH | 6.602 | 16.09% | 9.179 | 18.18% | 12.853 | 20.02% | 20.043 | 0.23% | 40.198 | 0.14% | 57.035 | 0.17% | 90.173 | 0.65% |
| ReEvo | 6.457 | 13.54% | 9.033 | 16.30% | 12.667 | 18.28% | N/A | N/A | N/A | N/A | N/A | N/A | N/A | N/A |
| HSEvo | 6.429 | 13.05% | 8.903 | 14.63% | 12.359 | 15.41% | N/A | N/A | N/A | N/A | N/A | N/A | N/A | N/A |
| MCTS-AHD | 6.358 | 11.80% | 8.839 | 13.80% | 12.403 | 15.82% | 20.035 | 0.27% | 40.206 | 0.12% | 57.020 | 0.20% | 89.061 | 1.88% |
| PathWise(Ours) | 6.245 | 9.81% | 8.758 | 12.76% | 12.276 | 14.63% | 20.046 | 0.21% | 40.216 | 0.09% | 57.082 | 0.09% | 90.719 | 0.05% |
| LLM-based AHD: *GPT-5-nano* (reasoning: low) | | | | | | | | | | | | | | |
| Funsearch | 6.389 | 12.35% | 8.941 | 15.12% | 12.701 | 18.60% | 20.037 | 0.26% | 40.173 | 0.20% | 57.035 | 0.17% | 90.609 | 0.17% |
| EoH | 6.380 | 12.19% | 8.920 | 14.85% | 12.541 | 17.11% | 20.043 | 0.23% | 40.182 | 0.18% | 57.046 | 0.15% | 90.609 | 0.17% |
| ReEvo | 7.287 | 28.13% | 10.115 | 30.23% | 14.083 | 31.51% | N/A | N/A | N/A | N/A | N/A | N/A | N/A | N/A |
| HSEvo | 6.346 | 11.59% | 8.792 | 13.20% | 12.223 | 14.14% | N/A | N/A | N/A | N/A | N/A | N/A | N/A | N/A |
| MCTS-AHD | 6.383 | 12.24% | 8.814 | 13.48% | 12.254 | 14.43% | 20.042 | 0.23% | 40.215 | 0.10% | 57.081 | 0.09% | 90.651 | 0.12% |
| PathWise(Ours) | 6.202 | 9.06% | 8.620 | 10.98% | 12.132 | 13.29% | 20.044 | 0.22% | 40.217 | 0.09% | 57.081 | 0.09% | 90.724 | 0.04% |
| LLM-based AHD: *GPT-5-nano* (reasoning: medium) | | | | | | | | | | | | | | |
| Funsearch | 6.321 | 11.15% | 8.761 | 12.80% | 12.159 | 13.54% | 20.039 | 0.25% | 40.190 | 0.16% | 57.041 | 0.16% | 90.627 | 0.15% |
| EoH | 6.354 | 11.73% | 8.809 | 13.41% | 12.222 | 14.12% | 20.045 | 0.22% | 40.215 | 0.10% | 57.051 | 0.14% | 90.654 | 0.12% |
| ReEvo | 6.284 | 10.50% | 8.848 | 13.92% | 12.463 | 16.38% | N/A | N/A | N/A | N/A | N/A | N/A | N/A | N/A |
| HSEvo | 6.274 | 10.32% | 8.744 | 12.58% | 12.178 | 13.72% | N/A | N/A | N/A | N/A | N/A | N/A | N/A | N/A |
| MCTS-AHD | 6.238 | 9.69% | 8.694 | 11.94% | 12.148 | 13.44% | 20.043 | 0.23% | 40.212 | 0.10% | 57.080 | 0.09% | 90.674 | 0.10% |
| PathWise(Ours) | 6.165 | 8.41% | 8.687 | 11.85% | **12.009** | **12.14%** | **20.073** | **0.08%** | **40.239** | **0.04%** | **57.109** | **0.04%** | **90.742** | **0.02%** |

step. Define $\mathcal{R} = (V_{t'} \cup V_{t'}^{\text{disc}}) \setminus \mathcal{F}$, where $V_{t'}^{\text{disc}}$ contains all nodes evaluated but not entailed during the inner loop. Thus, $\mathcal{R}$ consists of nodes that are not leaves of $G_{t'}$.

Population management prioritizes $\mathcal{F}$ based on performance:

$$\mathcal{P}_{r+1} = \begin{cases} \text{Top}_{N_p}(\mathcal{F}) & \text{if } |\mathcal{F}| \geq N_p, \\ \mathcal{F} \cup \text{Top}_{N_p - |\mathcal{F}|}(\mathcal{R}) & \text{otherwise.} \end{cases} \quad (8)$$

By carrying forward both high-performing heuristics and the structurally informative leaf nodes, the outer loop forms the next population from the best individuals while retaining the contextual structure captured during entailment.

# 4. Experiments

We evaluate the performance of PathWise on complex optimization tasks, focusing on NP-hard COPs. For our experimental evaluation, PathWise is applied to design heuristic functions within three search frameworks: the step-by-step constructive framework, Ant Colony Optimization (ACO), and Guided Local Search (GLS). Detailed framework definitions and configurations are provided in Appendix C. Our code is publicly available at https://github.com/oguzhangungordu/PathWise.

**Benchmarks.** PathWise is evaluated across diverse problem domains and search spaces using the following benchmarks: Traveling Salesman Problem (TSP), Knapsack Problem (KP), Capacitated Vehicle Routing Problem (CVRP),

Multiple Knapsack Problem (MKP), Orienteering Problem (OP), and Bin Packing Problem (BPP; including both offline and online variants). Details of the problem definitions and dataset construction procedures are provided in Appendix B.

**Experimental Settings.** Parameters of PathWise balance search depth and exploration. In experiments, we set $N_a = 2$, $N_w = 2$, $N_p = 6$, and $I_{\max} = 3$. We use GPT-5-nano[1] with low and medium reasoning levels, and GPT-4o-mini as a non-reasoning model. The sampling temperature for all agents is set to 1.0. Details of training/test sets and further experimental settings are in Appendix D.

**Baselines.** PathWise is evaluated against a comprehensive set of state-of-the-art baselines, categorized by their underlying search frameworks. For the step-by-step construction framework, we compare against manually designed Nearest-Greedy constructive heuristics and POMO (Kwon et al., 2020), a representative Neural Combinatorial Optimization (NCO) method. In the online BPP task, performance is measured against classic heuristics, specifically Best Fit and First Fit (Seiden, 2002). Within the ACO framework, we benchmark against the original ACO (Dorigo et al., 2006) and its neural-enhanced variant, DeepACO (Ye et al., 2023). For the GLS framework, we compare against Knowledge-guided Local Search (KGLS) (Arnold & Sörensen, 2019) and several iterative NCO solvers, including VRP-DACT

---

[1] GPT-5-nano(low) and GPT-5-nano(medium) denote different reasoning levels of the same base model.

*Table 2.* Designing heuristics with the ACO search framework for solving TSP, CVRP, MKP, OP, and offline BPP. Each test set contains 250 instances for TSP and CVRP, and 100 instances for the remaining problems. Performance of LLM-based AHD methods is averaged over 3 runs. Gaps are computed relative to the best-performing solver within each test set.

| Task | TSP | | | | CVRP | | | | MKP | | | | OP | | | | Offline BPP | | | |
|---|---|---|---|---|---|---|---|---|---|---|---|---|---|---|---|---|---|---|---|---|
| Test sets | $N{=}50$ | | $N{=}100$ | | $N{=}50, C{=}50$ | | $N{=}100, C{=}50$ | | $N{=}100, m{=}5$ | | $N{=}300, m{=}5$ | | $N{=}50$ | | $N{=}200$ | | $N{=}500, C{=}150$ | | $N{=}1000, C{=}150$ | |
| Methods | Obj.↓ | Gap | Obj.↓ | Gap | Obj.↓ | Gap | Obj.↓ | Gap | Obj.↑ | Gap | Obj.↑ | Gap | Obj.↑ | Gap | Obj.↑ | Gap | Obj.↓ | Gap | Obj.↓ | Gap |
| ACO | 6.143 | 6.19% | 9.191 | 12.66% | 13.358 | 50.77% | 22.587 | 50.32% | 21.258 | 3.59% | 53.640 | 7.77% | 14.128 | 6.63% | 49.556 | 8.43% | 210.670 | 3.40% | 420.630 | 3.73% |
| DeepACO | **5.785** | **0.00%** | **8.158** | **0.00%** | **8.860** | **0.00%** | **15.026** | **0.00%** | 21.649 | 1.82% | 56.250 | 3.28% | **15.132** | **0.00%** | 53.609 | 0.94% | **203.740** | **0.00%** | **405.490** | **0.00%** |
| *LLM-based AHD: GPT-4o-mini* | | | | | | | | | | | | | | | | | | | | |
| EoH | 5.892 | 1.85% | 8.382 | 2.75% | 9.451 | 6.67% | 16.796 | 11.78% | 21.884 | 0.75% | 56.740 | 2.44% | 14.792 | 2.25% | 53.180 | 1.74% | 207.388 | 1.79% | 414.585 | 2.24% |
| ReEvo | 5.896 | 1.92% | 8.351 | 2.37% | 9.484 | 7.04% | 16.749 | 11.47% | 22.024 | 0.12% | 57.051 | 1.91% | 14.760 | 2.46% | 51.602 | 4.65% | 206.787 | 1.50% | 412.943 | 1.84% |
| HSEvo | 5.927 | 2.45% | 8.361 | 2.49% | 9.836 | 11.02% | 17.112 | 13.88% | 21.838 | 0.96% | 56.672 | 2.56% | 14.839 | 1.94% | 53.678 | 0.81% | 206.010 | 1.11% | 411.145 | 1.39% |
| MCTS-AHD | 5.908 | 2.13% | 8.494 | 4.12% | 9.915 | 11.91% | 17.260 | 14.87% | 21.900 | 0.68% | 56.992 | 2.01% | 14.810 | 2.13% | 52.366 | 3.24% | 205.255 | 0.74% | 409.410 | 0.97% |
| PathWise(Ours) | 5.874 | 1.54% | 8.333 | 2.15% | 9.350 | 5.53% | 16.574 | 10.30% | 22.037 | 0.06% | 57.879 | 0.48% | 14.915 | 1.43% | **54.119** | **0.00%** | 205.100 | 0.67% | 409.590 | 1.01% |
| *LLM-based AHD: GPT-5-nano (reasoning: low)* | | | | | | | | | | | | | | | | | | | | |
| EoH | 5.967 | 3.14% | 8.685 | 6.46% | 10.226 | 15.42% | 17.696 | 17.77% | 21.886 | 0.74% | 56.682 | 2.54% | 14.589 | 3.59% | 50.210 | 7.21% | 205.087 | 0.66% | 408.707 | 0.79% |
| ReEvo | 6.148 | 6.27% | 9.352 | 14.64% | 10.669 | 20.42% | 18.141 | 20.73% | 21.870 | 0.82% | 56.215 | 3.34% | 14.662 | 3.11% | 49.209 | 9.07% | 205.610 | 0.92% | 410.330 | 1.19% |
| HSEvo | 5.979 | 3.35% | 8.910 | 9.22% | 10.676 | 20.50% | 18.188 | 21.04% | 21.909 | 0.64% | 56.774 | 2.38% | 14.614 | 3.42% | 50.507 | 6.67% | 205.577 | 0.90% | 409.793 | 1.06% |
| MCTS-AHD | 5.930 | 2.51% | 8.588 | 5.27% | 10.763 | 21.48% | 18.518 | 23.24% | 21.874 | 0.80% | 56.683 | 2.54% | 14.639 | 3.26% | 49.979 | 7.65% | 206.590 | 1.40% | 412.747 | 1.79% |
| PathWise(Ours) | 5.859 | 1.28% | 8.313 | 1.90% | 9.648 | 8.89% | 16.515 | 9.91% | 22.026 | 0.11% | 58.083 | 0.13% | 14.682 | 2.97% | 52.208 | 3.53% | 204.360 | 0.30% | 407.837 | 0.58% |
| *LLM-based AHD: GPT-5-nano (reasoning: medium)* | | | | | | | | | | | | | | | | | | | | |
| EoH | 5.945 | 2.76% | 8.638 | 5.88% | 9.605 | 8.41% | 16.886 | 12.38% | 21.973 | 0.35% | 57.378 | 1.34% | 14.578 | 3.66% | 51.048 | 5.66% | 205.248 | 0.74% | 409.788 | 1.06% |
| ReEvo | 6.047 | 4.53% | 8.935 | 9.52% | 10.034 | 13.25% | 17.226 | 14.64% | 21.808 | 1.10% | 55.983 | 3.74% | 14.686 | 2.95% | 48.511 | 10.36% | 205.510 | 0.87% | 410.177 | 1.16% |
| HSEvo | 6.050 | 4.58% | 9.105 | 11.61% | 10.552 | 19.10% | 17.998 | 19.78% | 21.803 | 1.12% | 56.119 | 3.51% | 14.457 | 4.46% | 40.910 | 24.41% | 206.155 | 1.19% | 411.725 | 1.54% |
| MCTS-AHD | 6.026 | 4.17% | 8.764 | 7.43% | 9.619 | 8.57% | 16.908 | 12.52% | 22.015 | 0.16% | 57.404 | 1.30% | 14.672 | 3.04% | 49.465 | 8.60% | 205.725 | 0.97% | 410.610 | 1.26% |
| PathWise(Ours) | 5.812 | 0.47% | 8.309 | 1.85% | 9.637 | 8.77% | 16.646 | 10.78% | **22.050** | **0.00%** | **58.159** | **0.00%** | 14.795 | 2.23% | 54.035 | 0.16% | 204.885 | 0.56% | 408.535 | 0.75% |

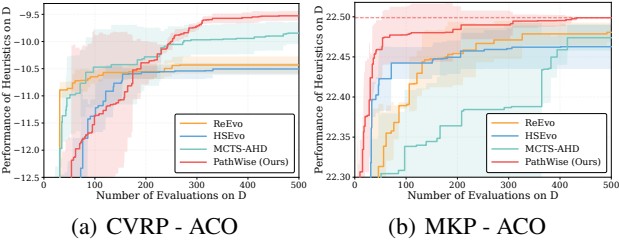

(a) CVRP - ACO  (b) MKP - ACO

*Figure 3.* Evolution curves of LLM-based AHD methods using `GPT-5-nano (low)` on (a) CVRP and `GPT-5-nano (medium)` on (b) MKP, with a limit of $n_e = 500$ heuristic evaluations. Each curve is averaged over the 3 runs used in Table 2.

(Ma et al., 2021), NeuOpt (Ma et al., 2023), NeuralGLS (Sui et al., 2023), and GNNGLS (Hudson et al., 2022).

PathWise is also benchmarked against leading open-source LLM-based AHD methods: FunSearch (Romera-Paredes et al., 2023), EoH (Liu et al., 2024a), ReEvo (Ye et al., 2024), HSEvo (Dat et al., 2025), and MCTS-AHD (Zheng et al., 2025). For LLM-based AHD methods, we report the mean performance over three independent runs with a limit of $n_e = 500$ heuristic evaluations per task and a 60-second execution time per heuristic on the training dataset $\mathcal{D}_{\text{train}}$.

### 4.1. Overall Results

The overall performance of PathWise on the test sets $\mathcal{D}_{\text{test}}$ is shown in Table 1 for the step-by-step construction framework and Table 2 for the ACO framework, with results

for the GLS framework provided in Appendix E. Additional cost analysis and output examples are provided in Appendix H and Appendix G, respectively.

**Step-by-Step Construction Framework.** The step-by-step construction (constructive) framework builds a solution incrementally by selecting one feasible component at a time. At each step, the heuristic evaluates available choices based on the partial solution and selects the next component (Pacheco-Valencia et al., 2019; Asani et al., 2023). Within this framework, AHD focuses on generating heuristic functions that prioritize candidate components during construction. Experiments are conducted on TSP and KP, with additional results on online BPP reported in Appendix E.

Table 1 shows that PathWise consistently outperforms the manually designed Greedy Construct heuristics and all LLM-based AHD methods on both TSP and KP across all test sets, showing that the improvements over baselines hold for both in-domain (ID) and out-of-domain (OOD) test sets and remain stable across different LLM backbones. Furthermore, performance improves systematically with model capability, with optimality gaps decreasing when moving from GPT-4o-mini to GPT-5-nano and with higher reasoning levels. To quantify improvements, we calculate the *mean relative gap improvement* (MRGI) of PathWise over existing LLM-based AHD baselines, obtained by averaging the relative gap improvements of PathWise across LLM-based methods for each test set and LLM backbone (see Appendix E for details). On TSP, PathWise achieves an overall average improvement of 20.38%, averaged across test sets and

LLM backbones, with improvement of 31.82% observed for GPT-5-nano (low). Similarly, on KP, PathWise achieves average improvements of 49.89% (GPT-4o-mini), 20.26% (GPT-5-nano (low)), and 65.20% (GPT-5-nano (medium)). PathWise further exhibits strong scalability with respect to problem size. When averaging the MRGI of each test set across LLM backbones, performance increases from 31.67% on the ID test set ($N=100$, $W=25$) to 81.34% on the largest OOD test set ($N=500$, $W=25$), indicating that the effectiveness grows as problem complexity increases. Moreover, PathWise achieves better performance on TSP ($N=200$) and KP test sets than the neural solver POMO, even though POMO requires task-specific training.

**Ant Colony Optimization Framework.** ACO is a population-based framework inspired by the collective foraging behavior of ants, where multiple ants construct solutions by repeatedly choosing the next move according to both accumulated search experience and heuristic guidance (Kim et al., 2025; Abir et al., 2025). Within this framework, AHD focuses on generating heuristic functions that guide ants' local decisions. Experiments are conducted TSP, CVRP, MKP, OP, and offline BPP, with extended results on a wider range of problem instances provided in Appendix E.

Table 2 shows that PathWise outperforms LLM-based AHD methods on 5 CO problems in both 4 ID test sets and 4 OOD test sets across all LLM backbones. PathWise achieves overall average MRGI gains of 60.22% on TSP, 38.73% on CVRP, 89.35% on MKP, 54.81% on OP, and 44.86% on offline BPP, averaged across test sets and LLM backbones within each problem. Furthermore, PathWise demonstrates strong scalability; for example, the average MRGI on OP across LLM backbones increases from 25.40% on the ID test set ($N=50$) to 84.22% on the OOD test set ($N=200$), consistent with the trends in the constructive framework. PathWise also consistently outperforms manually designed ACO heuristics across all test sets and LLM backbones. Moreover, PathWise achieves better performance on OP ($N=200$) and MKP than the neural solver DeepACO, even though DeepACO requires specialized per-scale training.

To evaluate heuristic evolution efficiency and stability, we compare best-so-far heuristic performance as a function of evaluation number. As shown in Figure 3, PathWise exhibits faster convergence and lower variance across tasks and LLM backbones. Notably, this behavior persists even when allowing twice the evaluation budget for LLM-based AHD baselines, as shown in Figure 1 (averaged over 5 runs).

### 4.2. Ablation Study

We conduct an ablation study on the TSP constructive task to evaluate the impact of core components in PathWise, using `GPT-5-nano(low)`, with results averaged over 5 runs. These experiments examine how planning through a world model, critic-driven feedback, and prompt-level diversity jointly improve heuristic discovery and generalizability. Parameter ablations are provided in Appendix F.4.

*Table 3.* Ablations on critic agents in *PathWise*. We report optimality gaps (%) on training (*TSP50*) and validation sets (*TSP20*, *TSP50*) for the step-by-step construction framework on TSP.

| Methods | *TSP20* | *TSP50* | *TSP50* |
|---|---|---|---|
| PathWise | 6.08% | 9.72% | 8.79% |
| *w/o* Policy Critic | 8.51% | 11.24% | 10.21% |
| *w/o* World Model Critic | 7.23% | 10.44% | 9.63% |
| *w/o* Policy Critic & World Model Critic | 11.73% | 14.85% | 14.42% |

**Contribution of Policy and World Model Critics.** To assess the impact of critic agents in PathWise, we selectively remove the policy critic and the world model critic with other components fixed. As shown in Table 3, removing either critic consistently degrades performance, indicating that critic feedback is central to effective heuristic evolution. Among the two, removing the policy critic results in a larger performance drop than removing the world model critic, highlighting the importance of feedback in guiding parent selection and operator choices. The world model critic provides complementary benefits by refining code-level modifications, often simplifying algorithmic structure or reducing time complexity, which improves execution efficiency and stabilizes training. Removing both critics causes a substantial drop, demonstrating their complementary roles in maintaining collaboration between the policy and world model over the entailment graph.

*Table 4.* Ablations on prompt perturbation and state shuffling in *PathWise*. We report optimality gaps and the Selection Diversity Rate (SDR) on training (*TSP50*) and validation sets (*TSP20*, *TSP50*) for the step-by-step construction framework on TSP.

| Methods | *TSP20* Gap (%) | *TSP50* Gap (%) | *TSP50* Gap (%) | SDR (%) |
|---|---|---|---|---|
| PathWise | 6.08% | 9.72% | 8.79% | 75.79% |
| *w/o* Prompt Perturbation & State Shuffling | 9.19% | 11.98% | 11.43% | 53.76% |
| *w/o* Prompt Perturbation | 8.52% | 10.15% | 9.58% | 70.30% |
| *w/o* State Shuffling | 6.08% | 9.93% | 9.02% | 61.03% |

**Effect of Prompt-level Diversity Mechanisms.** To evaluate prompt-level diversity in PathWise, we ablate prompt perturbation and state shuffling with all other components fixed. As shown in Table 4, removing both causes the largest performance drop and sharply reduces the Selection Diversity Rate (SDR), indicating that prompt-level diversity is critical for effective exploration. SDR measures the diversity of parent sets selected by the policy agent across entailment steps. Specifically, it is the ratio of the number of unique parent selections to the total number of policy selections over the evolutionary run, reflecting how broadly the policy explores the state. Among the two, removing prompt perturbation results in a larger performance drop, showing that

semantic variation in sampled actions is the primary driver of exploration. In contrast, removing state shuffling has a smaller effect on solution quality but significantly reduces SDR, suggesting it mainly mitigates positional bias and improves selection diversity rather than directly affecting heuristic quality. Higher SDR correlates with improved performance, indicating that selection diversity is essential for preventing premature convergence. These results suggest prompt-level diversity mechanisms enhance performance by increasing action and rollout contrast, yielding more informative feedback for heuristic evolution.

## 5. Conclusion

In this paper, we propose PathWise, a novel hybrid graph-based and population-based framework formulating AHD as state-aware planning through reasoning over an entailment graph, enabling reasoning about heuristic derivation over time. Through collaboration of policy, world model, and critic agents, PathWise supports structured, memory-aware, and diverse heuristic generation guided by past decisions and outcomes. Experiments show PathWise discovers better heuristics with fast convergence and strong generalizability.

## Acknowledgements

This work is supported in part by the DARPA SciFy program, Award No. HR001125C0302.

## Impact Statement

This paper presents work whose goal is to advance the field of Machine Learning. There are many potential societal consequences of our work, none which we feel must be specifically highlighted here.

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

# A. Related Work

## A.1. AHD and Hyper-Heuristics

Heuristic algorithms are central to solving COPs in domains such as logistics, scheduling, and design, where exact methods are often impractical due to NP-hardness (Kieffer et al., 2020). Consequently, metaheuristic approaches such as simulated annealing, tabu search, and iterated local search have been widely adopted to obtain high-quality solutions under limited computational budgets (Kirkpatrick et al., 1983; Glover, 1990; Lourenço et al., 2003; Desale et al., 2015; Tam et al., 2024). Despite their effectiveness, heuristic development has relied on domain expertise and manual trial-and-error, resulting in solvers that are costly to develop and generalize poorly across problem variants (Pillay & Qu, 2018; Choong et al., 2018).

AHD and hyper-heuristics aim to reduce this reliance on manual design by shifting the search from solutions to heuristics themselves (Burke et al., 2013). Early hyper-heuristic frameworks introduced the idea of selecting or generating heuristics at a higher level, either by choosing among predefined low-level heuristics or by constructing new heuristics from reusable components (Cowling et al., 2001; Sabar et al., 2013). A common realization of this paradigm is GP, where heuristics are represented as executable programs and evolved through mutation and recombination (Langdon & Poli, 2013). GP-based AHD has produced competitive heuristics for satisfiability, scheduling, routing, and packing problems (Fukunaga, 2002; Branke et al., 2016; Duflo et al., 2019). In parallel, learning-based hyper-heuristics model heuristic selection as a sequential decision problem and apply RL to adaptively choose heuristics during search, showing improved performance over static selection strategies in several optimization settings (Zhang et al., 2022; Dokeroglu et al., 2024).

## A.2. Evolutionary Algorithms with LLMs

Evolutionary algorithms (EAs) are search and optimization methods inspired by biological mechanisms such as natural selection, mutation, and recombination. They provide a general framework for search processes, where a population of candidates is iteratively improved through variation and selection (Bäck et al., 1997; Eiben & Smith, 2015). Within this framework, LLMs are typically used as generative operators that produce new candidates conditioned on selected parents, while fitness evaluation and population updates are handled externally following standard EAs. Several recent works adopt this paradigm by prompting LLMs to perform mutation- and crossover-like transformations on existing candidates, leveraging the LLM's ability to generate code or structured text rather than operating on fixed symbolic representations (Lehman et al., 2024; Meyerson et al., 2024; Lange et al., 2024). The resulting evolutionary loop maintains a population, selects parents based on performance, and uses the LLM to generate offspring that modify and combine prior solutions.

EAs with LLMs have been applied to program synthesis and algorithmic code generation, where candidate programs are refined over generations by iteratively improving promising implementations (Chen et al., 2023; Hemberg et al., 2024; Bradley et al., 2024). Other approaches apply evolutionary search to natural language outputs, evolving textual solutions under task-specific evaluation criteria, which shows that EA principles extend naturally to purely linguistic search spaces (Fernando et al., 2024; Xu et al., 2024). Closely related methods focus on evolving prompts themselves, using evolutionary operators to discover prompts that consistently elicit higher-quality outputs (Guo et al., 2024b; Grießhaber et al., 2025). Overall, EAs provide a practical way to structure LLM-based search by handling evaluation and selection externally while using LLMs for candidate generation (Wu et al., 2025).

## A.3. LLM-based AHD

Prior to LLM-based AHD, LLM-as-optimizers have been used for COPs, where the model directly improves solutions for individual problem instances rather than discovering reusable algorithms. In this paradigm, LLMs generate candidate solutions through in-context learning and iteratively refine them by conditioning on previously generated high-quality solutions together with their objective values (Yang et al., 2024a; Guo et al., 2024a; Liu et al., 2024b). This optimization process typically follows a loop in which an initial set of feasible solutions is produced for a given instance, the best-performing solutions are retained, and the LLM is repeatedly prompted to propose improved candidates that are evaluated and fed back into the context. While effective for improving solution quality on individual instances, LLM-as-optimizer methods face challenges on problems with large or complex search spaces, including limited exploration capability and sensitivity to context design (Nasir et al., 2024; Zhao et al., 2023; Liu et al., 2025b).

Moving beyond instance-level optimization, LLM-based AHD methods integrate LLMs into evolutionary search to automatically generate heuristics with minimal human intervention (Liu et al., 2023). Many approaches adopt population-based evolutionary procedures, where LLMs iteratively refine a population of candidate algorithms using evolutionary search

(Chen et al., 2023; Meyerson et al., 2024). Early methods such as FunSearch (Romera-Paredes et al., 2023) and EoH (Liu et al., 2024a) employ LLM-guided operators to evolve high-performing heuristics, enabling efficient exploration of large heuristic spaces. ReEvo (Ye et al., 2024) further incorporates a reflection mechanism to analyze generated heuristics and guide subsequent search steps (Shinn et al., 2023). Building on this, HSEvo (Dat et al., 2025) introduces diversity-aware population management and harmony search to promote population diversity and mitigate premature convergence (Shi et al., 2013). Beyond population-based approaches, tree-based methods have also been explored (Wang et al., 2025a). In particular, MCTS-AHD (Zheng et al., 2025) integrates LLMs with Monte Carlo Tree Search, applying the UCT algorithm to guide selection and expansion of heuristic nodes in a tree structure, enabling more structured exploration of the heuristic space (Świechowski et al., 2023).

Several LLM-based AHD works extend beyond single-heuristic, single-objective settings. (Liu et al., 2025c) focuses on generating a small set of complementary heuristics to improve coverage across diverse instance distributions; however, it does not learn or predict which heuristic to apply to a given instance at inference time, and thus operates at a set-level evaluation. (Yao et al., 2025) formulates heuristic generation as a multi-objective optimization problem, incorporating efficiency criteria in addition to heuristic performance. In PathWise and the baselines considered in this paper, heuristics are evaluated under the same search framework, sharing the same heuristic space and fitness landscape, with a focus on heuristic performance using an explicit instance-level inference procedure.

### A.4. LLM for Reasoning and Code Generation

LLMs have demonstrated capabilities in complex logical reasoning, mathematical problem-solving, and code generation (Yang et al., 2024b). However, standard chain-of-thought prompting often becomes unreliable on multi-step tasks due to error accumulation and limited long-horizon consistency, motivating structured reasoning approaches that make intermediate state, verification, and search more explicit (Yao et al., 2023; Li et al., 2023). Reasoning can be formulated as structure-aware planning with world models for symbolic state tracking and validation (Hao et al., 2023; Xiong et al., 2025b; 2026b), or combined with programmatic tools and formal representations to decompose subproblems and verify intermediate outputs (Yang et al., 2024c). Exploring multiple reasoning trajectories and aggregating their outcomes further mitigates failures associated with single forward-pass inference (Xiong et al., 2025a; 2026a).

Recently, learning from feedback and experience has emerged as a way to improve LLM reasoning, by evolving the context provided to the model across attempts and tasks (Gao et al., 2025). Iterative critique-and-revision mechanisms allow models to generate feedback on their own outputs and refine them over multiple passes (Shinn et al., 2023; Madaan et al., 2023). Other approaches treat prompts and instructions as objects of search, using performance feedback and textual edits to construct more effective and reusable contexts (Zhou et al., 2023b; Pryzant et al., 2023; Xiang et al., 2025; Zhang et al., 2025). In parallel, memory-based methods store, retrieve, and update information from prior interactions so that agents can reuse effective reasoning patterns and workflows across tasks, rather than relearning them each time (Liang et al., 2025; Ouyang et al., 2025; Chhikara et al., 2025; Xu et al., 2025). Together, these directions indicate that improving reasoning and code generation depends not only on better prompting for individual instances, but also on mechanisms that accumulate feedback and experience into reusable context over time (Gao et al., 2025). These frameworks generally are evaluated on tasks such as question answering or tool use, which focus on solving individual problem instances. In contrast, AHD introduces additional challenges, as it requires learning executable algorithms that generalize across instance distributions and evolve under long-horizon heuristic discovery. PathWise addresses these challenges by applying learning from feedback and experience to the heuristic discovery setting, using state-aware reasoning over an entailment graph that provides a stateful representation of derivation history and structured reasoning to guide heuristic evolution over time.

## B. Details of Benchmark Problems

This section presents the problem definitions, mathematical formulations, and the procedures used for benchmark instance generation. Experiments are conducted on a set of representative NP-hard COPs, including the Traveling Salesman Problem (TSP), Knapsack Problem (KP), Capacitated Vehicle Routing Problem (CVRP), Multiple Knapsack Problem (MKP), Orienteering Problem (OP), and Bin Packing Problem (BPP). For BPP, both offline and online settings are considered.

### B.1. Traveling Salesman Problem (TSP)

**Definition.** The Traveling Salesman Problem (TSP) seeks a minimum-length Hamiltonian tour that visits each city exactly once and returns to the starting city.

**Formulation.** Let $G = (V, E)$ be a complete graph with $V = \{1, \ldots, n\}$ and edge costs $c_{ij} \geq 0$. Let $x_{ij} \in \{0, 1\}$.

$$
\begin{aligned}
\text{minimize} \quad & \sum_{i \in V} \sum_{j \in V} c_{ij} x_{ij}, \\
\text{s.t.} \quad & \sum_{j \in V} x_{ij} = 1, \ \sum_{i \in V} x_{ij} = 1, \ \sum_{i \in S} \sum_{j \in S} x_{ij} \leq |S| - 1, \ x_{ij} \in \{0, 1\},
\end{aligned}
\tag{9}
$$

where $\forall i, j \in V$ and $\forall S \subset V, \ 2 \leq |S| \leq n - 1$.

**Instance generation.** Node coordinates are uniformly sampled from $[0, 1]^2$. The distance matrix is computed using Euclidean distances, with diagonal elements set to 1 to prevent self-loops, and a small constant ($10^{-5}$) added to diagonal to prevent numerical issues in the GLS framework.

### B.2. Knapsack Problem (KP)

**Definition.** The Knapsack Problem (KP) aims to select a subset of items with maximum total value subject to a single weight constraint.

**Formulation.** Given values $v_i$, weights $w_i$, and weight $W$, let $x_i \in \{0, 1\}$.

$$
\begin{aligned}
\text{maximize} \quad & \sum_{i=1}^{n} v_i x_i, \\
\text{s.t.} \quad & \sum_{i=1}^{n} w_i x_i \leq W, \quad x_i \in \{0, 1\}.
\end{aligned}
\tag{10}
$$

**Instance generation.** Item weights and values are uniformly sampled from $[0, 1]$. The knapsack weight capacity is set to 25 for all problem sizes, except 12.5 for the 50-item instances, following the settings of ReEvo (Ye et al., 2024).

### B.3. Capacitated Vehicle Routing Problem (CVRP)

**Definition.** The Capacitated Vehicle Routing Problem (CVRP) seeks a set of minimum-cost vehicle routes originating and ending at a depot, such that all customer demands are satisfied without exceeding vehicle capacity.

**Formulation.** Let $G = (V, E)$ include depot 0, customers $V \setminus \{0\}$, demands $d_i$, capacity $C$, and $x_{ij} \in \{0, 1\}$ (Toth & Vigo, 2002).

$$
\begin{aligned}
\text{minimize} \quad & \sum_{i \in V} \sum_{j \in V} c_{ij} x_{ij}, \\
\text{s.t.} \quad & \sum_{j \in V} x_{ij} = 1, \ \sum_{i \in V} x_{ij} = 1, \ \sum_{i \in S} d_i \leq C \sum_{i \in S} \sum_{j \notin S} x_{ij}, \ x_{ij} \in \{0, 1\},
\end{aligned}
\tag{11}
$$

where $\forall i, j \in V \setminus \{0\}$ and $\forall S \subseteq V \setminus \{0\}$.

**Instance generation.** Node coordinates are uniformly sampled from $[0, 1]^2$. A depot is fixed at coordinates $[0.5, 0.5]$. Customer demands are uniformly sampled from $\{1, 2, \ldots, 9\}$. The vehicle capacity is set to 50, following the settings of ReEvo (Ye et al., 2024).

### B.4. Multiple Knapsack Problem (MKP)

**Definition.** The Multiple Knapsack Problem (MKP) generalizes the knapsack problem to multiple knapsacks, where items must be assigned to at most one knapsack.

**Formulation.** Let $v_i$ be the value of item $i$, $w_{ij}$ be the weight of item $i$ in knapsack $j$, $C_j$ be the capacity of knapsack $j$, and

$x_{ij} \in \{0, 1\}$ indicate whether item $i$ is assigned to knapsack $j$ (Pisinger, 1999).

$$
\text{maximize} \quad \sum_{i=1}^{n} \sum_{j=1}^{m} v_i x_{ij},
$$
$$
\text{s.t.} \quad \sum_{i=1}^{n} w_{ij} x_{ij} \leq C_j, \ \sum_{j=1}^{m} x_{ij} \leq 1, \ x_{ij} \in \{0, 1\}. \tag{12}
$$

**Instance generation.** For MKP with $n$ items and $m = 5$ knapsacks, item values $v_i$ are uniformly sampled from $[0, 1]$. The weight $w_{ij}$ of item $i$ in knapsack $j$ is uniformly sampled from $[0, 1]$. For each knapsack $j$, the capacity $C_j$ is uniformly sampled from $[\max_i w_{ij}, \sum_i w_{ij}]$, then all weights are normalized as $w_{ij} \leftarrow w_{ij}/C_j$ such that $C_j = 1$ after normalization, ensuring well-defined instances, following the settings of ReEvo (Ye et al., 2024).

### B.5. Orienteering Problem (OP)

**Definition.** The Orienteering Problem (OP) seeks a path that maximizes collected rewards from visited nodes while satisfying a total travel budget constraint.

**Formulation.** Let rewards be $r_i$, travel costs be $c_{ij}$, and the budget be $B$ (Vansteenwegen et al., 2011).

$$
\text{maximize} \quad \sum_{i \in V} r_i x_i,
$$
$$
\text{s.t.} \quad \sum_{(i,j) \in E} c_{ij} y_{ij} \leq B, \quad x_i \in \{0, 1\}. \tag{13}
$$

**Instance generation.** Node coordinates are uniformly sampled from $[0, 1]^2$. The travel costs $c_{ij}$ are computed as Euclidean distances between nodes. Prize values are computed as $r_i = (1 + \lfloor 99 \cdot d_i/d_{\max} \rfloor)/100$ where $d_i$ is the Euclidean distance from node $i$ to the depot (node 0), normalized by the maximum distance $d_{\max}$, and then normalized such that $\max_i r_i = 1$, following the settings of ReEvo (Ye et al., 2024). The travel budget $B$ varies by problem size: $B = 3.0$ for $n = 50$, $B = 4.0$ for $n = 100$, $B = 5.0$ for $n = 200$, $B = 6.0$ for $n = 300$, and $B = 7.0$ for larger instances (Kool et al., 2019).

### B.6. Bin Packing Problem (BPP)

**Definition.** The Bin Packing Problem (BPP) aims to pack items of varying sizes into the minimum number of bins with fixed capacity. In the offline setting, all item sizes are known in advance, whereas in the online setting, items arrive sequentially and must be assigned to bins without knowledge of future items.

**Formulation.** Let item sizes be $s_i$ and bin capacity be $C$.

$$
\text{minimize} \quad \sum_j y_j,
$$
$$
\text{s.t.} \quad \sum_j x_{ij} = 1, \ \sum_i s_i x_{ij} \leq C y_j, \ x_{ij}, y_j \in \{0, 1\}. \tag{14}
$$

**Instance generation.** For the offline version, item demands are uniformly sampled from $\{20, 21, \ldots, 100\}$, following the protocol used in Funsearch (Romera-Paredes et al., 2023) and EoH (Liu et al., 2024a). The bin capacity is set to 150. For the online version, item sizes are sampled from a Weibull distribution with shape parameter 3 and scale parameter 45, then clipped to a maximum value of 100. The bin capacity is set to 100 or 500 depending on the evaluation setting (Levine & Ducatelle, 2004), following the protocol used in ReEvo (Ye et al., 2024) and MCTS-AHD (Zheng et al., 2025).

## C. Details of General Search Frameworks

This section provides detailed descriptions of the general search frameworks used in our experiments. These frameworks provide the algorithmic backbone within which our LLM-evolved heuristics operate. For each framework, we describe the underlying search mechanism and specify the particular heuristic functions that LLM-based AHD methods learn for different COPs.

### C.1. Step-by-Step Construction

**Framework Description.** Step-by-step construction (also known as greedy constructive heuristics) is a classical approach for solving COPs where solutions are built incrementally by making sequential decisions. Starting from an empty or partial solution, the algorithm iteratively selects the next component to add based on a selection rule until a complete feasible solution is constructed. This approach is characterized by its computational efficiency and ability to quickly generate reasonable solutions, though it typically cannot backtrack from previously made decisions.

**Search Procedure.** Let $S_t$ denote the partial solution at step $t$, and $C_t$ represent the set of feasible components that can be added to $S_t$. At each step, a selection function $h : S_t \times C_t \to \mathbb{R}$ assigns a score to each candidate component $c \in C_t$. The next component is selected as:

$$c_{t+1} = \arg\max_{c \in C_t} h(S_t, c) \tag{15}$$

The algorithm terminates when no more components can be added, yielding the final solution $S^* = S_T$.

**Problem-Specific Heuristics.** We apply this framework to three problems, where LLM-based methods evolve the selection function $h$:

- **TSP-Constructive**: Starting from a depot node, the algorithm iteratively selects the next city to visit until all cities are visited and the tour returns to the depot. The evolved heuristic $h$ determines which unvisited city should be selected next based on the current city, remaining unvisited cities, and the distance matrix.

- **KP-Constructive**: Items are sequentially selected for inclusion in the knapsack until no more items can fit within the capacity constraint. The evolved heuristic $h$ ranks items based on remaining capacity, item weights, and item values to determine the next item to select.

- **BPP-Online**: Items arrive sequentially and must be immediately assigned to bins without future knowledge. For each arriving item, the evolved heuristic $h$ computes a priority score for each bin based on the item size and bin remaining capacities, selecting the bin with the highest priority.

### C.2. Ant Colony Optimization

**Framework Description.** Ant Colony Optimization (ACO) is a population-based metaheuristic inspired by the foraging behavior of ants, originally proposed by Dorigo et al. (Dorigo et al., 2006). The algorithm maintains a probabilistic model in the form of pheromone trails $\tau_{ij}$ defined over solution components (e.g., edges in routing problems), which guide the stochastic construction of solutions by a population of artificial ants. At each construction step, ants sample solution components according to a transition rule that combines pheromone information with heuristic measures, typically controlled by weighting parameters. Through repeated solution construction and pheromone update cycles, ACO balances exploration and exploitation, progressively reinforcing high-quality components while evaporating inferior ones to improve solution quality over time (Jardee & Sheppard, 2025).

**Search Procedure.** Each ant $k$ constructs a solution by probabilistically selecting components based on pheromone levels $\tau_{ij}$ and heuristic information $\eta_{ij}$. The probability of selecting component $j$ from component $i$ is:

$$p_{ij}^k = \frac{[\tau_{ij}]^\alpha \cdot [\eta_{ij}]^\beta}{\sum_{l \in \mathcal{N}_i^k} [\tau_{il}]^\alpha \cdot [\eta_{il}]^\beta} \tag{16}$$

where $\mathcal{N}_i^k$ is the set of feasible components for ant $k$ at component $i$, and $\alpha, \beta$ are parameters controlling the relative importance of pheromone and heuristic information.

After all ants construct solutions with costs $L^k$, pheromones are updated as:

$$\tau_{ij} \leftarrow (1 - \rho)\tau_{ij} + \sum_{k=1}^{m} \Delta\tau_{ij}^k, \quad \text{where} \quad \Delta\tau_{ij}^k = \begin{cases} 1/L^k & \text{if ant } k \text{ uses component } (i, j) \\ 0 & \text{otherwise} \end{cases} \tag{17}$$

where $\rho \in (0, 1)$ is the evaporation rate and $m$ is the number of ants.

**Problem-Specific Heuristics.** The heuristic information matrix $\eta_{ij}$ plays a crucial role in guiding solution construction. We apply ACO to five problems where LLM-based methods evolve problem-specific heuristics:

- **TSP-ACO**: The heuristic $\eta_{ij}$ provides edge-level guidance for including edge $(i, j)$ in the tour. The evolved heuristic takes the distance matrix as input and returns a matrix indicating how promising each edge is for tour construction.

- **CVRP-ACO**: For vehicle routing with capacity constraints, the heuristic $\eta_{ij}$ incorporates both distance information and customer demands to guide route construction. The evolved heuristic considers the distance matrix, node coordinates, customer demands, and vehicle capacity to produce edge-level guidance.

- **MKP-ACO**: The heuristic $\eta_i$ provides item-level guidance indicating how promising each item is for inclusion. The evolved heuristic takes item prizes and the multi-dimensional weight matrix as input, returning a vector of heuristic values for items, considering the normalized capacity constraints.

- **OP-ACO**: The heuristic $\eta_{ij}$ balances prize collection and distance constraints. The evolved heuristic takes prize values, the distance matrix, and the maximum path length budget as input, producing an edge-level heuristic matrix that guides path construction toward high-value nodes while respecting the travel budget.

- **BPP-Offline-ACO**: For offline bin packing, the heuristic $\eta_{ij}$ indicates how promising it is to pack items $i$ and $j$ in the same bin. The evolved heuristic takes item demands and bin capacity as input, returning a pairwise affinity matrix that guides the grouping of items into bins.

### C.3. Guided Local Search (GLS)

**Framework Description.** Guided Local Search (GLS) is a metaheuristic that enhances local search by strategically escaping local optima through solution modification (Voudouris & Tsang, 1999). The algorithm maintains penalty weights on solution features (e.g., edges in TSP) that dynamically adjust during the search. When local search converges to a local optimum, GLS penalizes features that contribute to the current solution's cost, effectively modifying the search landscape to encourage exploration of different regions. This mechanism combines the efficiency of local search with the ability to escape local optima through adaptive penalties. In our experiments, the GLS-family framework is instantiated as Knowledge-Guided Local Search (KGLS) (Arnold & Sörensen, 2019), which injects a learned knowledge matrix to guide when and how features are penalized and how the current solution is perturbed. This design follows prior LLM-based AHD work that evolves penalty/knowledge heuristics for GLS and deploys them within KGLS (Ye et al., 2024; Zheng et al., 2025).

**Search Procedure.** Let $s_t$ denote the current solution at iteration $t$, $f(s)$ the original objective function, and $p_i$ the penalty associated with feature $i$ (e.g., an edge in TSP). GLS considers an augmented objective:

$$g(s) = f(s) + \lambda \sum_{i \in I(s)} p_i, \tag{18}$$

where $I(s)$ is the set of features present in solution $s$, and $\lambda$ controls the penalty strength (typically set as $\lambda = \alpha \cdot f(s^*)/n$, where $s^*$ is the best solution found, $n$ is the problem size, and $\alpha$ is a scaling parameter).

When local search reaches a local optimum under $g(\cdot)$, penalties are updated using a utility score. In KGLS, the utility is modulated by a learned knowledge matrix $\eta$:

$$\text{util}(i, j) = \frac{\eta_{ij} \cdot c_{ij}}{1 + p_{ij}}, \tag{19}$$

where $\eta_{ij}$ is the learned knowledge signal for feature (edge) $(i, j)$, and $c_{ij}$ is the cost contribution of feature $(i, j)$ (e.g., edge distance $d_{ij}$ for edge $(i, j)$ in TSP). Features with maximal utility are identified and penalized: $p_{ij} \leftarrow p_{ij} + 1$. This penalty update modifies the augmented objective $g(s)$, and local search resumes under the updated landscape.

KGLS alternates between two phases: (i) Local Improvement, where classical local search operators (e.g., 2-opt and relocate for TSP) are applied to improve the current solution under the augmented objective $g(s)$; and (ii) Guided Perturbation, where upon convergence to a local optimum, features with high utility scores (computed using the learned knowledge matrix) are penalized to modify the search landscape. The algorithm maintains and updates the best solution $s^*$ encountered throughout the search, returning it upon termination.

**Problem-Specific Heuristic.** We apply the GLS framework to TSP, where the LLM-based methods evolve the knowledge matrix that guides the penalty mechanism:

- **TSP-GLS**: Edges are penalized when they appear in local optima. The evolved heuristic takes the distance matrix as input and returns a knowledge matrix $\eta \in \mathbb{R}^{n \times n}$, where $\eta_{ij}$ encodes distance-weighted knowledge about how problematic or undesirable edge $(i, j)$ is for solution quality. When local search converges, the utility of each edge $(i, j)$ in the current tour is computed as $\text{util}(i, j) = \eta_{ij}/(1 + p_{ij})$, where $p_{ij}$ is the accumulated penalty on edge $(i, j)$. Edges with maximum utility are then penalized to guide perturbation. This allows the search to prioritize penalizing edges identified as problematic by the learned heuristic while accounting for how frequently they have already been penalized, leading to more effective escape from local optima.

## D. Experimental Details

This section provides the experimental configuration used across reported results, including dataset construction, LLM-based AHD settings, NCO baselines, and the hyperparameters of the underlying search frameworks.

### D.1. Dataset Configuration

The datasets used in the experiments are organized according to the underlying search framework and problem type. For each framework–problem pair, a fixed training dataset $\mathcal{D}_{\text{train}}$ and a separate test dataset $\mathcal{D}_{\text{test}}$ are constructed. The instance size (e.g., number of nodes, items, or customers) and the number of instances for both datasets are reported in Table 5.

During heuristic design, all LLM-based AHD methods are evaluated only on the training set $\mathcal{D}_{\text{train}}$, which represents the in-distribution data used to guide the search through performance feedback. Once the stopping criterion is met, defined by reaching the heuristic evaluation budget $n_e$, the final heuristic produced by each method is evaluated on the held-out test set $\mathcal{D}_{\text{test}}$, which is used to assess out-of-distribution generalization. For each problem domain, $\mathcal{D}_{\text{train}}$ and $\mathcal{D}_{\text{test}}$ are synthesized by sampling random instances under fixed settings, following ReEvo (Ye et al., 2024), EoH (Liu et al., 2024a), and MCTS-AHD (Zheng et al., 2025). Compared to these prior works, we evaluate on test sets with larger instance sizes and a greater number of instances, in order to assess performance under more challenging and diverse problem settings (Smith-Miles et al., 2014; Dunning et al., 2018; Smith-Miles et al., 2021; Sim et al., 2025).

*Table 5.* Benchmark dataset configuration across frameworks and problems.

| Framework | Problem | $\mathcal{D}_{\text{train}}$ | | $\mathcal{D}_{\text{test}}$ | |
| --- | --- | --- | --- | --- | --- |
| | | Size | #Instances | Size | #Instances |
| | TSP | 50 | 64 | {50,100,200} | 250 |
| Constructive | KP | 100 | 64 | {50,100,200,500} | 250 |
| | Online BPP | {1k,5k} | 4 | {1k,5k,10k} | 10 |
| | TSP | 50 | 5 | {50,100} | 250 |
| | CVRP | 50 | 10 | {50,100} | 250 |
| ACO | MKP | 100 | 5 | {100,200,300,500,1k} | 100 |
| | OP | 50 | 10 | {50,100,200,500} | 100 |
| | Offline BPP | 500 | 5 | {500,1000} | 100 |
| GLS | TSP | 200 | 10 | {100,200,500,1k} | 250 |

### D.2. LLM-Based AHD Methods Configuration

Details of the evaluation budget, language model choices, and settings used by LLM-based AHD methods are provided below.

**Evaluation Budget** $n_e$ For all LLM-based AHD methods, the maximum number of heuristic evaluations is fixed to $n_e = 500$ for all problem domains. A uniform evaluation budget is used to ensure fair comparison across methods and to attribute performance differences to the quality of the search strategy rather than to unequal computational effort. Although increasing $n_e$ enables broader exploration, overly large evaluation budgets can mask methodological differences due to the stochastic nature of LLMs. With sufficiently many sampled heuristics, improvements may primarily reflect the effects of generating many random candidate heuristics rather than consistent search guidance or convergence behavior (Zhao et al., 2025; Wang et al., 2025b). By setting $n_e$ to a moderate value, the evaluation emphasizes a method's ability to efficiently guide heuristic generation and achieve reliable improvements within a bounded computational budget, rather than relying on large-scale sampling. In addition, all heuristic evaluations are executed on a single AMD Ryzen Threadripper PRO 7985WX CPU, and each method's total training time is capped at 6 hours per run.

**Large Language Models Configuration.** Both non-reasoning and reasoning LLMs are evaluated to study how models with different reasoning capabilities behave in AHD. The pre-trained LLM is *gpt-4o-mini-2024-07-18* for `GPT-4o-mini` and `GPT-5-nano` for the GPT-5 family, which provides a fast and cost-efficient reasoning-capable model with strong performance on reasoning benchmarks involving multi-step inference and algorithmic problem solving (Srivastava et al., 2025; Singh et al., 2025). For PathWise, the temperature is fixed to 1.0 for all agents. For other LLM-based AHD methods, temperature settings follow their original configurations; for example, ReEvo increases the temperature to 0.3 during the initialization phase to promote diverse heuristic sampling. An exception is `GPT-5-nano`, for which the temperature is fixed to 1.0 as it is not configurable. Experiments with `GPT-5-nano` are conducted using low verbosity and two reasoning levels (low and medium).

**Baselines Configuration** We follow the original algorithmic configurations for all baseline LLM-based AHD methods (e.g., number of parents in operators, operator ordering, mutation rate, number of islands, and harmony search hyperparameters). For ReEvo and HSEvo, the population size is set to 30 during the initialization stage and reduced to 10 in subsequent stages. For EoH, the population size is set to 20 for Online BPP and to 10 for all other tasks.

ReEvo and HSEvo require seed heuristic functions to initialize the search and maintain sufficient variation among candidate heuristics. When all individuals in the population converge to the same objective value, no further optimization is possible and the search terminates. Following the setup in MCTS-AHD (Zheng et al., 2025), we use the seed functions proposed in (Ye et al., 2024) for ACO and GLS frameworks, random selection heuristics for constructive TSP, and the best-known heuristic from (Romera-Paredes et al., 2023) for Online BPP.

Although HSEvo does not originally report results for certain problem–framework combinations, including CVRP-ACO, MKP-ACO, TSP-ACO, and constructive settings, its architecture closely follows ReEvo with additional diversity mechanisms. We therefore include these configurations to provide a comprehensive comparison. For some tasks, such as constructive KP for ReEvo and HSEvo, results are not reported due to early termination caused by convergence of the heuristics in the population to identical objective values (Zheng et al., 2025).

### D.3. NCO Methods Configuration

For the step-by-step constructive framework, we report results of POMO (Kwon et al., 2020) on TSP and KP, where solutions are constructed sequentially using a learned constructive policy. POMO solutions are generated using a single start without test-time augmentation. Within the ACO framework, we compare against DeepACO (Ye et al., 2023) on TSP, CVRP, MKP, OP, and Offline BPP, where neural networks provide learned heuristic guidance for ant decision rules and are trained separately for each dataset size. Following DeepACO (Ye et al., 2023), the number of steps per epoch is set to 128 and the number of training epochs is set to 5, while the number of ants (ant population size) and graph sparsification parameters are chosen according to problem-specific settings across different instance sizes. ACO results are generated using the baseline heuristic configurations described in (Ye et al., 2023). For the GLS framework, we include iterative NCO solvers—VRP-DACT (Ma et al., 2021), NeuOpt (Ma et al., 2023), NeuralGLS (Sui et al., 2023), and GNNGLS (Hudson et al., 2022)—and report their performance on TSP instances, where the maximum number of operations applied to a solution is fixed to $T = 1200$ for VRP-DACT and NeuOpt.

### D.4. Search Frameworks Configuration

Table 6 details the search framework hyperparameters used for heuristic evaluations during heuristic evolution. These configurations specify the population size for ACO, perturbation moves for GLS, and the total number of iterations for both frameworks.

*Table 6.* Search framework hyperparameters used during heuristic evolution across problems.

| Framework | Problem | Search Hyperparameters | | |
| --- | --- | --- | --- | --- |
| | | Population Size (#Ants) | Perturbation Moves | #Iterations |
| GLS | TSP | — | 30 | 1200 |
| ACO | TSP | 30 | — | 100 |
| | CVRP | 30 | — | 100 |
| | MKP | 10 | — | 50 |
| | OP | 20 | — | 50 |
| | Offline BPP | 20 | — | 15 |

# E. Extended Results

This section presents evaluation metric details and extended experimental results. We report additional results for the step-by-step construction framework on online BPP in Table 7, results for the GLS framework on TSP in Table 8, extended ACO results on larger-size MKP and OP instances in Table 9, results on TSPLIB instances in Table 10, results with the open-source LLM backbone DeepSeek-V3.2 in Table 11, and statistical significance testing on TSP-ACO and KP-Constructive in Tables 12 and 13. Together, these results provide a more comprehensive view of PathWise's performance across different search frameworks, problem variants, problem sizes, and LLM backbones.

**Evaluation Metric.** To quantify performance improvements over LLM-based AHD baseline methods, let $\text{Gap}_m$ denote the optimality gap of a baseline method $m$ on a given test set, and let $\text{Gap}_{\text{PW}}$ denote the corresponding gap achieved by PathWise. The *relative gap improvement* (RGI) of PathWise over method $m$ is defined as

$$\text{RGI}(m) = \frac{\text{Gap}_m - \text{Gap}_{\text{PW}}}{\text{Gap}_m} \times 100\%.$$

For a fixed test set and LLM backbone, we compute the *mean relative gap improvement* (MRGI) by averaging the RGI values of PathWise over all available LLM-based AHD baselines for that test set:

$$\text{MRGI} = \frac{1}{|\mathcal{M}|} \sum_{m \in \mathcal{M}} \text{RGI}(m).$$

where $\mathcal{M}$ denotes the set of LLM-based AHD methods included in the comparison for the corresponding CO problem. This yields an MRGI value for each combination of CO problem, test set, and LLM backbone. These MRGI values are used to compare the performance of PathWise across different LLM backbones and problem sizes. By averaging MRGI values over test sets of the same problem, we evaluate how performance differs between ID and OOD settings. Averaging MRGI across test sets for a fixed LLM backbone summarizes the effect of model capability, while averaging across LLM backbones reflects overall robustness independent of a specific model choice.

**Online BPP under the Step-by-Step Construction Framework.** In online BPP, the heuristic decides how each incoming item is assigned to a bin sequentially based on the current bin states. Table 7 reports results on 6 test sets covering both ID and OOD problem sizes. Across all LLM backbones, PathWise consistently achieves the lowest average gap to the lower bound when averaged over the 6 test sets. When comparing average performance, PathWise yields higher MRGI over existing LLM-based AHD baselines: $54.48\%$ under GPT-4o-mini, $33.86\%$ under GPT-5-nano (low), and $50.96\%$ under GPT-5-nano (medium).

*Table 7.* Designing step-by-step construction heuristics for online BPP. Performance gaps to the lower bound are reported, averaged over 3 runs for each LLM-based AHD method. Each test set consists of 10 Weibull BPP instances, with in-domain scales in $D_{test}$ underlined. Test-set scales are abbreviated (e.g., 1k_100 denotes 1,000 items with capacity $W=100$).

| Test sets | 1k_100 | 1k_500 | 5k_100 | 5k_500 | 10k_100 | 10k_500 | Avg. |
|---|---|---|---|---|---|---|---|
| Online BPP | | | | | | | |
| Best Fit | 4.73% | 0.25% | 4.19% | 0.50% | 3.99% | 0.45% | 2.35% |
| First Fit | 5.05% | 0.62% | 4.56% | 0.52% | 4.31% | 0.46% | 2.59% |
| LLM-based AHD: *GPT-4o-mini* | | | | | | | |
| EoH | 4.80% | 0.38% | 3.66% | 0.49% | 3.29% | 0.45% | 2.18% |
| ReEvo | 4.66% | 0.25% | 3.27% | 0.50% | 2.70% | 0.44% | 1.97% |
| HSEvo | 4.11% | 0.66% | 3.20% | 0.53% | 3.01% | 0.45% | 1.99% |
| MCTS-AHD | 4.79% | 0.25% | 4.15% | 0.48% | 4.01% | 0.45% | 2.36% |
| PathWise(Ours) | 2.46% | 0.29% | 1.38% | 0.38% | 1.23% | 0.33% | **1.01%** |
| LLM-based AHD: *GPT-5-nano* (reasoning: low) | | | | | | | |
| EoH | 3.38% | 0.37% | 2.04% | 0.74% | 1.82% | 0.63% | 1.50% |
| ReEvo | 4.70% | 0.25% | 4.20% | 0.47% | 4.02% | 0.45% | 2.35% |
| HSEvo | 3.96% | 0.52% | 1.77% | 0.55% | 1.28% | 0.51% | 1.43% |
| MCTS-AHD | 4.36% | 0.44% | 3.93% | 0.50% | 3.77% | 0.45% | 2.24% |
| PathWise(Ours) | 3.10% | 0.37% | 2.11% | 0.54% | 1.65% | 0.48% | **1.38%** |
| LLM-based AHD: *GPT-5-nano* (reasoning: medium) | | | | | | | |
| EoH | 4.27% | 0.25% | 2.32% | 0.50% | 1.95% | 0.62% | 1.65% |
| ReEvo | 3.22% | 0.37% | 2.62% | 0.47% | 2.52% | 0.42% | 1.60% |
| HSEvo | 4.42% | 0.48% | 3.70% | 0.50% | 3.49% | 0.45% | 2.17% |
| MCTS-AHD | 3.19% | 0.29% | 2.09% | 0.39% | 1.91% | 0.35% | 1.37% |
| PathWise(Ours) | 2.92% | 0.29% | 0.92% | 0.26% | 0.84% | 0.23% | **0.91%** |

**TSP under the GLS Framework.** Table 8 reports optimality gaps for designing heuristics within the GLS framework on TSP. Performance improves systematically with model capability, with gaps generally decreasing when moving from GPT-4o-mini to GPT-5-nano (medium). In terms of performance comparison among LLM-based AHD methods, PathWise achieves average MRGI values of 3.77% under GPT-4o-mini, 13.91% under GPT-5-nano (low), and 25.04% under GPT-5-nano (medium), where averages are computed across MRGI values of test sets for each LLM backbone.

*Table 8.* Designing heuristics within the GLS general framework for solving TSP. We report optimality gaps (%) to the optimum. Each LLM-based AHD method is run 3 times and we report average optimality gaps.

| TSP-GLS | | | | |
|---|---|---|---|---|
| $N =$ | 100 | 200 | 500 | 1,000 |
| Optimal | 0.0000% | 0.0000% | 0.0000% | 0.0000% |
| KGLS | **0.0034%** | 0.2270% | 0.9578% | 1.5348% |
| NCO methods with the GLS general framework | | | | |
| VRP-DACT | 1.7943% | 91.9267% | N/A | N/A |
| NeuOpt | 0.2950% | 0.9152% | N/A | N/A |
| NeuralGLS | 0.470% | 3.622% | N/A | N/A |
| GNNGLS | 0.705% | 3.522% | N/A | N/A |
| LLM-based AHD: *GPT-4o-mini* | | | | |
| ReEvo | 0.0068% | 0.1981% | 0.9969% | 1.5974% |
| HSEvo | 0.0059% | 0.2008% | 0.9810% | 1.6097% |
| MCTS-AHD | 0.0078% | 0.2053% | 1.0094% | 1.5929% |
| PathWise(Ours) | 0.0060% | 0.1919% | 1.0009% | 1.6021% |
| LLM-based AHD: *GPT-5-nano* (reasoning: low) | | | | |
| ReEvo | 0.0060% | **0.1815%** | 0.9699% | 1.5947% |
| HSEvo | 0.0122% | 0.2185% | 0.9558% | 1.5568% |
| MCTS-AHD | 0.0134% | 0.2414% | 0.9352% | 1.4960% |
| PathWise(Ours) | 0.0052% | 0.2015% | 0.9295% | 1.4745% |
| LLM-based AHD: *GPT-5-nano* (reasoning: medium) | | | | |
| ReEvo | 0.0110% | 0.2169% | 0.9825% | 1.6103% |
| HSEvo | 0.0154% | 0.2132% | 0.9273% | 1.4930% |
| MCTS-AHD | 0.0113% | 0.2228% | 0.9950% | 1.6128% |
| PathWise(Ours) | 0.0036% | 0.1896% | **0.9090%** | **1.4039%** |

**MKP and OP under the ACO Framework on Larger-Size Instances.** Table 9 shows extended ACO results for MKP and OP on a wider range of instance sizes. Across both problems, PathWise consistently outperforms existing LLM-based AHD methods across all problem sizes and LLM backbones. On MKP, PathWise achieves the best performance on all evaluated test sets and surpasses DeepACO under GPT-5-nano (medium). On OP, PathWise surpasses DeepACO on the OOD $N$=200 test set under GPT-4o-mini. These results demonstrate that PathWise remains effective at larger scales and can match or exceed DeepACO, which is a task-specific neural solver baseline.

*Table 9.* Designing heuristics with the ACO general framework for solving MKP and OP. Each test set contains 100 instances, and the performance of LLM-based AHD methods is averaged over 3 runs. Gaps are computed relative to the best-performing solver within each test set.

| Task | MKP ($m = 5$) | | | | | | | | | | OP | | | | | | | |
|---|---|---|---|---|---|---|---|---|---|---|---|---|---|---|---|---|---|---|
| Test sets | $N$=100 | | $N$=200 | | $N$=300 | | $N$=500 | | $N$=1000 | | $N$=50 | | $N$=100 | | $N$=200 | | $N$=500 | |
| Methods | Obj.↑ | Gap | Obj.↑ | Gap | Obj.↑ | Gap | Obj.↑ | Gap | Obj.↑ | Gap | Obj.↑ | Gap | Obj.↑ | Gap | Obj.↑ | Gap | Obj.↑ | Gap |
| ACO | 21.258 | 3.59% | 41.446 | 1.74% | 53.640 | 7.77% | 100.986 | 3.77% | 183.450 | 5.15% | 14.128 | 6.63% | 29.199 | 3.82% | 49.556 | 8.43% | 107.870 | 12.46% |
| DeepACO | 21.649 | 1.82% | 41.980 | 0.47% | 56.250 | 3.28% | 102.535 | 2.29% | 186.550 | 3.55% | **15.132** | **0.00%** | **30.358** | **0.00%** | 53.609 | 0.94% | **123.218** | **0.00%** |
| LLM-based AHD: *GPT-4o-mini* | | | | | | | | | | | | | | | | | | |
| EoH | 21.884 | 0.75% | 41.463 | 1.70% | 56.740 | 2.44% | 101.949 | 2.85% | 186.275 | 3.69% | 14.792 | 2.25% | 30.002 | 1.17% | 53.180 | 1.74% | 116.719 | 5.27% |
| ReEvo | 22.024 | 0.12% | 41.724 | 1.08% | 57.051 | 1.91% | 102.057 | 2.75% | 184.660 | 4.53% | 14.760 | 2.46% | 29.285 | 3.53% | 51.602 | 4.65% | 106.049 | 13.93% |
| HSEvo | 21.838 | 0.96% | 41.358 | 1.94% | 56.672 | 2.56% | 101.814 | 2.98% | 185.874 | 3.90% | 14.839 | 1.94% | 30.089 | 0.89% | 53.678 | 0.81% | 118.600 | 3.75% |
| MCTS-AHD | 21.900 | 0.68% | 41.544 | 1.50% | 56.992 | 2.01% | 102.416 | 2.41% | 187.024 | 3.30% | 14.810 | 2.13% | 30.042 | 1.04% | 52.366 | 3.24% | 115.665 | 6.13% |
| PathWise(Ours) | 22.037 | 0.06% | 42.043 | 0.32% | 57.879 | 0.48% | 104.378 | 0.54% | 192.138 | 0.66% | 14.915 | 1.43% | 30.283 | 0.25% | **54.119** | **0.00%** | 118.971 | 3.45% |
| LLM-based AHD: *GPT-5-nano* (reasoning: low) | | | | | | | | | | | | | | | | | | |
| EoH | 21.886 | 0.74% | 41.386 | 1.87% | 56.682 | 2.54% | 101.835 | 2.96% | 185.788 | 3.94% | 14.589 | 3.59% | 29.274 | 3.57% | 50.210 | 7.21% | 93.210 | 24.35% |
| ReEvo | 21.870 | 0.82% | 41.300 | 2.08% | 56.215 | 3.34% | 101.075 | 3.68% | 180.189 | 6.84% | 14.662 | 3.11% | 28.760 | 5.26% | 49.209 | 9.07% | 99.252 | 19.45% |
| HSEvo | 21.909 | 0.64% | 41.478 | 1.66% | 56.774 | 2.38% | 102.056 | 2.75% | 186.665 | 3.49% | 14.614 | 3.42% | 28.771 | 5.23% | 50.507 | 6.67% | 108.405 | 12.02% |
| MCTS-AHD | 21.874 | 0.80% | 41.443 | 1.74% | 56.683 | 2.54% | 101.802 | 2.99% | 185.246 | 4.22% | 14.639 | 3.26% | 28.734 | 5.35% | 49.979 | 7.65% | 107.064 | 13.11% |
| PathWise(Ours) | 22.026 | 0.11% | 42.150 | 0.07% | 58.083 | 0.13% | 104.829 | 0.11% | 193.093 | 0.17% | 14.682 | 2.97% | 29.716 | 2.11% | 52.208 | 3.53% | 113.932 | 7.54% |
| LLM-based AHD: *GPT-5-nano* (reasoning: medium) | | | | | | | | | | | | | | | | | | |
| EoH | 21.973 | 0.35% | 41.631 | 1.30% | 57.378 | 1.34% | 102.706 | 2.13% | 185.393 | 4.15% | 14.578 | 3.66% | 29.438 | 3.02% | 51.048 | 5.66% | 114.316 | 7.22% |
| ReEvo | 21.808 | 1.10% | 40.997 | 2.80% | 55.983 | 3.74% | 100.287 | 4.44% | 181.652 | 6.08% | 14.686 | 2.95% | 28.572 | 5.88% | 48.511 | 10.36% | 105.295 | 14.55% |
| HSEvo | 21.803 | 1.12% | 41.066 | 2.64% | 56.119 | 3.51% | 100.563 | 4.17% | 182.750 | 5.51% | 14.457 | 4.46% | 26.977 | 11.14% | 40.910 | 24.41% | 57.929 | 52.99% |
| MCTS-AHD | 22.015 | 0.16% | 41.832 | 0.82% | 57.404 | 1.30% | 104.154 | 0.75% | 191.478 | 1.00% | 14.672 | 3.04% | 28.922 | 4.73% | 49.465 | 8.60% | 111.069 | 9.86% |
| PathWise(Ours) | **22.050** | **0.00%** | **42.178** | **0.00%** | **58.159** | **0.00%** | **104.942** | **0.00%** | **193.416** | **0.00%** | 14.795 | 2.23% | 30.005 | 1.16% | 54.035 | 0.16% | 120.198 | 2.45% |

**Comparison of LLM-based AHD Methods on TSPLIB.** Evaluation is conducted on real-world TSP benchmarks from the TSPLIB dataset (Reinelt, 1991). Table 10 compares LLM-based AHD methods on TSPLIB instances using the step-by-step construction framework. Each heuristic is executed three times with different starting nodes, and optimality gaps are computed from the averaged objective values. Across instances, PathWise achieves the lowest average optimality gap and attains the best result on the majority of TSPLIB instances.

*Table 10.* Results of LLM-based AHD methods for the TSP on TSPLIB instances using a step-by-step construction framework. For each instance, heuristics are run 3 times with different starting nodes, and the reported optimality gap is computed from the averaged results. The best result per instance is highlighted in bold.

| Instance | ReEvo | HSEvo | MCTS-AHD | PathWise |
|---|---|---|---|---|
| ts225.tsp | 20.60% | 13.38% | **5.57%** | 15.31% |
| rat99.tsp | 14.65% | 17.02% | 16.48% | **10.63%** |
| bier127.tsp | 9.46% | 16.85% | 14.83% | **8.15%** |
| lin318.tsp | 21.31% | **13.01%** | 17.63% | 15.48% |
| eil51.tsp | 13.10% | **6.52%** | 9.42% | 12.32% |
| d493.tsp | 18.24% | 13.84% | 12.74% | **12.47%** |
| kroB100.tsp | 12.28% | 15.41% | **10.35%** | 12.43% |
| kroC100.tsp | 15.33% | 9.95% | 12.97% | **9.82%** |
| ch130.tsp | 19.74% | 11.59% | **9.60%** | 10.55% |
| pr299.tsp | 24.34% | 17.23% | 20.59% | **12.53%** |
| fl417.tsp | 27.57% | 22.07% | 20.81% | **16.59%** |
| d657.tsp | 24.62% | 20.12% | **15.14%** | 15.63% |
| kroA150.tsp | 19.90% | 16.44% | 15.51% | **11.93%** |
| pr264.tsp | 17.87% | 18.49% | 19.84% | **15.40%** |
| pr226.tsp | 17.84% | 23.10% | 19.77% | **8.90%** |
| pr439.tsp | 21.95% | 23.03% | 17.96% | **13.65%** |
| Average Opt. Gap | 18.68% | 16.13% | 14.95% | **12.61%** |

**Evaluation with an Open-Source LLM Backbone.** To assess the dependence of PathWise on proprietary LLMs, we additionally evaluate it with the open-source model DeepSeek-V3.2 (Liu et al., 2025a) on TSP under the ACO framework and KP under the step-by-step construction framework, using the same evaluation protocol as in Section 4. As shown in Table 11, PathWise continues to outperform LLM-based AHD baselines across problems and test sets with this open-source backbone, indicating that the framework generalizes beyond closed-source models. MCTS-AHD is excluded from the TSP-ACO comparison due to its prohibitively long training time per run.

*Table 11.* Designing heuristics with the open-source DeepSeek-V3.2 model on TSP-ACO and KP-Constructive. We report mean optimality gaps (%). Entries marked "N/A" indicate that the method was not run for that task in this evaluation.

| Task | TSP-ACO | | KP-Constructive | | |
|---|---|---|---|---|---|
| Test sets | $N{=}50$ | $N{=}100$ | $N{=}50, W{=}12.5$ | $N{=}200, W{=}25$ | $N{=}500, W{=}25$ |
| LLM-based AHD: *DeepSeek-V3.2* | | | | | |
| ReEvo | 3.53% | 8.59% | N/A | N/A | N/A |
| HSEvo | 2.77% | 6.40% | N/A | N/A | N/A |
| MCTS-AHD | N/A | N/A | 0.25% | 0.09% | 0.06% |
| PathWise(Ours) | 1.86% | 3.78% | 0.21% | 0.07% | 0.04% |

**Statistical Significance Testing.** To assess the statistical significance of PathWise's improvements over LLM-based AHD baselines, we conduct 6 independent runs of PathWise and the baselines on TSP-ACO and KP-Constructive across multiple LLM backbones. For each test set, we report mean optimality gap, standard deviation (Std), and the p-value of a one-sided $t$-test comparing PathWise to each baseline. Tables 12 and 13 show that PathWise improvements over all baselines are statistically significant ($p < 0.05$) across problems, sizes, and LLM backbones. MCTS-AHD is excluded from the TSP-ACO comparison due to its prohibitively long training time per run.

*Table 12.* Statistical significance testing on TSP-ACO. Mean optimality gap (%), standard deviation (Std), and one-sided $t$-test p-values between each baseline and PathWise are reported over 6 runs.

| Test sets | $N{=}50$ | | | $N{=}100$ | | |
|---|---|---|---|---|---|---|
| Methods | Gap↓ | Std | $p$-value | Gap↓ | Std | $p$-value |
| LLM-based AHD: *GPT-5-nano* (reasoning: medium) | | | | | | |
| ReEvo | 7.70% | 3.62% | 0.0047 | 14.19% | 4.79% | 0.0015 |
| HSEvo | 4.77% | 0.69% | 0.0014 | 10.91% | 2.03% | 0.0006 |
| PathWise(Ours) | 1.93% | 1.41% | - | 4.67% | 2.62% | - |
| LLM-based AHD: *DeepSeek-V3.2* | | | | | | |
| ReEvo | 3.53% | 0.98% | 0.0034 | 8.59% | 2.96% | 0.0043 |
| HSEvo | 2.77% | 1.00% | 0.0403 | 6.40% | 2.74% | 0.0377 |
| PathWise(Ours) | 1.86% | 0.41% | - | 3.78% | 1.50% | - |

*Table 13.* Statistical significance testing on KP-Constructive. Mean optimality gap (%), standard deviation (Std), and one-sided $t$-test p-values between each baseline and PathWise are reported over 6 runs.

| Test sets | $N{=}50, W{=}12.5$ | | | $N{=}100, W{=}25$ | | |
|---|---|---|---|---|---|---|
| Methods | Gap↓ | Std | $p$-value | Gap↓ | Std | $p$-value |
| LLM-based AHD: *GPT-4o-mini* | | | | | | |
| MCTS-AHD | 0.2583% | 0.0133% | 0.0002 | 0.1117% | 0.0098% | 0.0096 |
| PathWise(Ours) | 0.2217% | 0.0075% | - | 0.0983% | 0.0041% | - |
| LLM-based AHD: *GPT-5-nano* (reasoning: medium) | | | | | | |
| MCTS-AHD | 0.2167% | 0.0151% | 0.0020 | 0.0900% | 0.0126% | 0.0026 |
| PathWise(Ours) | 0.1117% | 0.0538% | - | 0.0517% | 0.0214% | - |

# F. Further Methodological Details & Extended Ablation Studies

In this section, Appendix F.1 presents the prompt templates used by the policy, world model, and critic agents. Appendix F.2 lists the exploration phrase inventories used for prompt-level diversity. Appendix F.3 provides the algorithmic pseudocode summarizing the overall procedure. Appendix F.4 reports extended ablation studies on PathWise hyperparameters.

### F.1. Prompt Templates for Agents

Prompts used across different agents are provided below. The initialization prompt used to generate heuristics in the initial population is described first, followed by the complete prompt templates governing the Policy, World Model, and Critic agents. Each agent operates under a predefined system prompt defining its functional role, paired with a structured user prompt dynamically populated from the current search state. These templates map how the entailment graph, routed reflections, and heuristic evaluations condition agent behavior. The problem descriptions, function descriptions, function signatures, and task-specific contexts follow formats used in ReEvo (Ye et al., 2024).

---

**Initialization Prompt**

```
[SYSTEM PROMPT]
You are an expert in the domain of optimization heuristics.

-----------------------------------------------------------

[USER PROMPT]
Write a {function_name} function for {problem_description}

{function_description}

Function signature:
{function_signature}

Create a novel heuristic approach for this problem.

Format your response as:
```python
```

```
[your generated code here]
```

Description: [Algorithmic description of this heuristic's approach and key features less than 30 words]

Derivation Rationale: [Brief explanation of the reasoning behind this heuristic design and why it should perform well for this problem]

## Policy Agent Prompt

```
[SYSTEM PROMPT]
You are an expert in heuristic evolution and search strategy design. Your task is to
select parent(s) from the given candidates and provide a directive that guides the
generation of better heuristics.

-------------------------------------------------------------
[USER PROMPT]
You are controlling the evolutionary search for {problem_description}

{function_description}

I have k existing heuristics listed below. For each heuristic, I provide its ID,
description, the heuristics used to derive it with their descriptions and objective
values, its derivation logic, its full code, and its own objective value:

ID: # Heuristic identifier
Description: # High-level description of the heuristic idea
Heuristics used to derive this candidate:
# List of parent heuristics with their descriptions and objective values used to generate this heuristic
Derivation logic:
# Description of how this heuristic was constructed or modified (derivation rationale)
Code:
# Full Python implementation of the heuristic function
Objective value: # Objective value on the evaluation dataset

...

ID: # Heuristic identifier
Description: # High-level description of the heuristic idea
Heuristics used to derive this candidate:
# List of parent heuristics with their descriptions and objective values used to generate this heuristic
Derivation logic:
# Description of how this heuristic was constructed or modified (derivation rationale)
Code:
# Full Python implementation of the heuristic function
Objective value: # Objective value on the evaluation dataset

Reflection:
# Reflection generated by the Policy Critic Agent based on the outputs of the previous entailment step

Task:
Choose one or more parent(s) and propose a directive for deriving a new heuristic, based
on the reflection and current heuristic candidates. Your selected parent(s) and
directive should help guide the creation of heuristics that achieve lower objective values
in future generations. {exploratory_phrase}

You must choose parent ID(s) only from the k heuristics listed above.

Consider the following when choosing parent(s) and generating the directive:
- Objective values and performance trends
- Diversity of approaches in the current candidates
- Derivation history (what has been tried and what has not been explored yet)

Format your response as:
PARENTS: [list of ID(s) selected from current heuristic candidates]
DIRECTIVE: [novel instruction less than 2 sentences describing how to
            modify, extend, or invent new logic from the selected parent(s)]

Do not give additional explanations.
```

## World Model Agent Prompt

```
[SYSTEM PROMPT]
You are an expert in the domain of optimization heuristics.

-------------------------------------------------------------
[USER PROMPT]
Write a {function_name} function for {problem_description}

{function_description}

I have k existing algorithms with their codes and objective values as follows:

ID: # Parent heuristic identifier
# Parent heuristic description
# Parent heuristic full code
Objective value: # Objective value on the evaluation dataset

...

ID: # Parent heuristic identifier
# Parent heuristic description
# Parent heuristic full code
Objective value: # Objective value on the evaluation dataset

Directive:
# Derivation rationale generated by the Policy Agent describing how to
# modify, combine, or invent new logic from the parent heuristic(s)
{exploratory_phrase}

Reflection:
# Reflection generated by the World Model Critic Agent based on the outputs of the previous entailment step

Write an improved function {function_signature}_v2 that follows the directive and
reflection by keeping the same function signature. The new algorithm should have
an objective value lower than all algorithms.

Output algorithm description and code. First write: '''Description: <Algorithmic description of the heuristic's
approach using less than 30 words>'''. Then enclose your code with Python code block:'''python ...'''
```

## Policy Critic Prompt

```
[SYSTEM PROMPT]
You are an expert in analyzing heuristic evolution. Your task is to give hints for
how future parent choices and directive designs should improve the heuristic
evolution process.

-------------------------------------------------------------
[USER PROMPT]
Analyze heuristic evolution for {problem_description}

{function_description}

I have k existing heuristics listed below. For each heuristic, I provide its ID,
description, the heuristics used to derive it with their descriptions and objective
values, its derivation logic, its full code, and its own objective value:

ID: # Heuristic identifier
Description: # High-level description of the heuristic idea
Heuristics used to derive this candidate:
# List of parent heuristics with their descriptions and objective values used to generate this heuristic
Derivation logic:
# Description of how this heuristic was constructed or modified (derivation rationale)
Code:
# Full Python implementation of the heuristic function
Objective value: # Objective value on the evaluation dataset

...

ID: # Heuristic identifier
Description: # High-level description of the heuristic idea
Heuristics used to derive this candidate:
# List of parent heuristics with their descriptions and objective values used to generate this heuristic
Derivation logic:
# Description of how this heuristic was constructed or modified (derivation rationale)
Code:
# Full Python implementation of the heuristic function
```

```
Objective value: # Objective value on the evaluation dataset

Actions taken and their results:
Based on the heuristic candidates above, several evolution steps (called actions)
were performed. Each action consists of:
- a chosen parent set,
- a directive describing how to modify or combine those parents, and
- the rollouts generated from that directive.

The actions below are listed from best to worst by the average objective value of their rollouts.
Each rollout is one heuristic produced from an action, including its description and objective value.

Action 0
Parent IDs: # Parent IDs selected in this action
Directive:  # Directive (derivation rationale) used in this action
Rollouts:
  rollout_0: # Rollout description and objective value
  rollout_1: # Rollout description and objective value

...

Action N_a - 1
Parent IDs: # Parent IDs selected in this action
Directive:  # Directive (derivation rationale) used in this action
Rollouts:
  rollout_0: # Rollout description and objective value
  rollout_1: # Rollout description and objective value

Reflection Task:
1. Analyze the outcomes of the taken actions based on current heuristic candidates.
Identify patterns behind which actions performed best and which performed worst,
evaluate how their rollouts improved upon or became worse than their parent heuristics,
and diagnose the key reasons behind these shifts to ground your hints.
2. Correlate success with the parent selection strategies and directives above.

You respond with concise hints for both improving parent selection and directives toward lower
objective values.

Do not refer to specific parent IDs, rollout names, or code blocks.

Write your reflection using less than 60 words.
```

## World Model Critic Prompt

```
[SYSTEM PROMPT]
You are an expert in the domain of optimization heuristics. Your task is to give
hints for designing better heuristics.

------------------------------------------------------------
[USER PROMPT]
Below are two {function_signature} functions for {problem_description}

{function_description}

You are provided with two code versions below, where the second version performs better than the first one.

Worse code:
Description: # Description of the lower-performing heuristic
Objective value: # Objective value of the lower-performing heuristic
Code:
# Full Python implementation of the lower-performing heuristic

Better code:
Description: # Description of the higher-performing heuristic
Objective value: # Objective value of the higher-performing heuristic
Code:
# Full Python implementation of the higher-performing heuristic

You respond with some hints for designing better heuristics to achieve lower objective values,
based on the two code versions. Phrase your hints as design principles, not tied to names, paths,
or labels appearing in the code. Write less than 30 words.
```

## F.2. Exploration Phrase Inventories

Exploration phrase inventories used to perturb the prompts of the Policy and World Model agents during heuristic evolution are provided below. As shown in Algorithm 3, at each inner entailment step $t$, an exploration phrase is sampled from $\Phi_{\mathrm{p}}$ for the Policy Agent and from $\Phi_{\mathrm{wm}}$ for the World Model Agent with probability $\varepsilon(t)$. These phrases encourage novelty early in the search and gradually diminish as the exploration rate anneals, promoting behavioral diversity without modifying the entailment-graph state.

---

**Policy Agent Explorative Phrase Inventory ($\Phi_{\mathrm{p}}$)**

```
•Favor unusual parent combinations.
•Explore less common heuristic structures.
•Prioritize novelty in parent selection.
•Try a nonstandard way to blend parent logic.
•Choose parents that differ the most in style.
•Favor unconventional directives.
•Promote risky or experimental parent mixes.
•Favor out-of-pattern directive ideas.
•Choose parents with conflicting logic for innovation.
•Prioritize exploration over refinement for this step.
•Encourage a fresh angle on how parents are merged.
•Introduce a twist in the directive logic.
•Propose a directive unlike previously tried patterns.
•Inject a novel angle into how parents are combined.
•Favor creativity over safety in directive formation.
•Promote a directive that breaks common heuristic habits.
•Invent a directive that departs from usual conventions.
•Attempt to introduce more novel mechanisms.
```

---

**World Model Agent Explorative Phrase Inventory ($\Phi_{\mathrm{wm}}$)**

```
•Create a new algorithm that has a totally different form from the given algorithms.
•Try generating codes with different structures, flows or algorithms.
•Introduce a novel program segment that fundamentally changes the logic.
•Create new mechanisms or equations that have not appeared before.
•Attempt to introduce more novel mechanisms and new equations or programme segments.
•Modify the structure of the algorithm rather than refining existing parts.
•Develop a new pathway in the code that changes how decisions are made.
•Construct a new rule or formulation that improves the existing methods.
•Come up with an original computational idea that changes the overall procedure.
```

---

## F.3. Algorithmic Pseudocode

Algorithm 1 summarizes the overall **PathWise** procedure, which organizes heuristic search across outer population updates and inner entailment graph construction. Algorithm 2, ENTAILMENTGRAPHCONSTRUCTION, corresponds to the inner entailment graph construction shown in Figure 2(a). Algorithm 3, ENTAILMENTSTEP, specifies the multi-agent interaction underlying each inner update, aligning with Figure 2(b).

Across all algorithms, each node is represented as $v = (h, \kappa, d, P(h; \mathcal{D}), \mathrm{PM})$, where $h$ denotes executable heuristic code, $\kappa$ the derivation rationale, $d$ a compact algorithmic description, $P(h; \mathcal{D})$ performance on the training set, and $\mathrm{PM}$ the parent metadata. In the initial population $\mathcal{P}_0$, parent metadata is empty since no entailment has yet occurred; in later populations, root nodes inherit their stored fields, including PM when available.

---

**Algorithm 1** PathWise: Planning through World Model for Automated Heuristic Design via Self-Evolving LLMs

---

**Inputs:** Training instance set $\mathcal{D}$; performance $P(h; \mathcal{D})$; population size $N_p$; actions per step $N_a$; rollouts per action $N_w$; max inner entailment steps $I_{\max}$; max total evaluations $n_e$; linear schedule $(\varepsilon^{\text{init}}, \varepsilon^{\text{final}})$; phrase lists $\Phi_{\text{p}}$, $\Phi_{\text{wm}}$.
**Outputs:** Best overall heuristic $h^\star$; entailment graphs $\{G\}$ across outer iterations.

  1: Initialize agents $\boldsymbol{\pi}_{\mathbf{p}}, \boldsymbol{\pi}_{\mathbf{wm}}, \boldsymbol{\pi}_{\mathbf{p\_critic}}, \boldsymbol{\pi}_{\mathbf{wm\_critic}}$                  ▷ Instantiate policy, world model, and critic LLMs
  2: $r \leftarrow 0$;    $\ell \leftarrow 0$                                  ▷ $\ell$: evaluation counter over all rollouts
  3: Initialize population $\mathcal{P}_0$; evaluate $P(h; \mathcal{D})$ for each $h$ stored in nodes $v \in \mathcal{P}_0$
  4: $v^\star \leftarrow \arg\max\{ P(h; \mathcal{D}) \mid h \in \mathcal{P}_0 \}$             ▷ Initialize best overall node (implicitly includes $h^\star$)
  5: $\rho_p(0) \leftarrow \varnothing$;   $\rho_{wm}(0) \leftarrow \varnothing$                       ▷ Initial reflections are empty
  6: **while** $\ell < n_e$ **do**
  7:      \\ *Inner entailment graph construction for population $\mathcal{P}_r$*
  8:      $(G_{t'}, V_{t'}^{\text{disc}}, \rho_p, \rho_{wm}, \ell, v^\star) \leftarrow \text{ENTAILMENTGRAPHCONSTRUCTION}(\mathcal{P}_r, \rho_p, \rho_{wm}, \ell, v^\star)$
  9:      \\ *Population update (leaf-first)*
 10:     $\mathcal{F} \leftarrow \text{LeafNodes}(G_{t'})$               ▷ Leaf nodes of $G_{t'}$ that involved in at least one entailment step
 11:     $\mathcal{R} \leftarrow (V_{t'} \cup V_{t'}^{\text{disc}}) \setminus \mathcal{F}$      ▷ All evaluated nodes (entailed or non-entailed), excluding leaf nodes of $G_{t'}$
 12:     **if** $|\mathcal{F}| \geq N_p$ **then**
 13:         $\mathcal{P}_{r+1} \leftarrow \text{Top}_{N_p}(\mathcal{F})$                          ▷ Next population from best leaves
 14:     **else**
 15:         $\mathcal{P}_{r+1} \leftarrow \mathcal{F} \cup \text{Top}_{N_p - |\mathcal{F}|}(\mathcal{R})$                ▷ Fill remaining slots from best non-leaves
 16:     **end if**
 17:     $r \leftarrow r + 1$
 18: **end while**
 19: $h^\star \leftarrow$ heuristic code stored in $v^\star$
 20: **Output:** $h^\star$ and entailment graphs $\{G\}$

---

**Population Initialization and Management Details.** In the initial population construction, we prompt the LLM $5N_p$ times using the initialization prompt, evaluate all generated heuristics on $\mathcal{D}_{\text{train}}$, and form $\mathcal{P}_0$ by selecting the top $N_p$ heuristics with *distinct* performance values $P(h; \mathcal{D}_{\text{train}})$. This filtering avoids retaining multiple heuristics with identical evaluation outcomes arising from stochastic sampling. The same principle is applied during population management in later outer iterations: when the number of selected leaf nodes is insufficient, remaining population slots are filled by selecting non-leaf nodes with the highest performance values, again enforcing distinct performance values among selected heuristics. Together, these initialization and management rules promote a diverse set of heuristics in the population while preserving performance-based selection.

---

**Algorithm 2** ENTAILMENTGRAPHCONSTRUCTION: Inner Entailment Loop for One Population

---

**Inputs:** Population $\mathcal{P}_r$; reflections $\rho_p(0), \rho_{wm}(0)$; counter $\ell$; best overall node $v^\star$.
**Outputs:** Graph $G_{t'}$; non-entailed nodes set $V_{t'}^{\text{disc}}$; updated reflections; updated $\ell$ and $v^\star$.

  1: $V_0 \leftarrow \mathcal{P}_r$                    ▷ Population elements are root nodes of the entailment graph, $v = (h, \kappa, d, P(h; \mathcal{D}), \text{PM})$
  2: $E_0 \leftarrow \varnothing$;   $V_0^{\text{disc}} \leftarrow \varnothing$
  3: $s_0 \leftarrow V_0$                                       ▷ Initial state is the initial node set
  4: $t \leftarrow 0$
  5: **while** $(t < I_{\max}) \wedge (|s_t| > 1)$ **do**          ▷ Run until $I_{\max}$ steps or the frontier collapses to one node
  6:      \\ *Entailment step*
  7:      $(V_{t+1}, E_{t+1}, s_{t+1}, V_{t+1}^{\text{disc}}, \rho_p(t+1), \rho_{wm}(t+1), v^\star, \ell)$
  8:        $\leftarrow \text{ENTAILMENTSTEP}(V_t, E_t, s_t, V_t^{\text{disc}}, \rho_p(t), \rho_{wm}(t), v^\star, \ell)$
  9:     $t \leftarrow t + 1$
 10: **end while**
 11: $t' \leftarrow t$                                  ▷ Index of the final inner step for this outer iteration
 12: $G_{t'} \leftarrow (V_{t'}, E_{t'})$
 13: **return** $(G_{t'}, V_{t'}^{\text{disc}}, \rho_p(t'), \rho_{wm}(t'), \ell, v^\star)$

---

**Algorithm 3** ENTAILMENTSTEP: Multi-Agent Entailment Update

---

**Inputs:** $(V_t, E_t)$; state $s_t$; non-entailed nodes set $V_t^{\text{disc}}$; reflections $\rho_p(t), \rho_{wm}(t)$; best overall node $v^\star$; counter $\ell$.

**Outputs:** $(V_{t+1}, E_{t+1}, s_{t+1}, V_{t+1}^{\text{disc}})$; $\rho_p(t+1), \rho_{wm}(t+1)$; updated $v^\star$ and $\ell$.

1:  \\ *Exploration schedule*
2:  $\varepsilon(\ell) \leftarrow \varepsilon^{\text{init}} + (\varepsilon^{\text{final}} - \varepsilon^{\text{init}}) \cdot \ell/n_e$        ▷ Linear annealing with evaluation count
3:  \\ *State shuffling*
4:  Randomly permute the order of nodes in $s_t$ before prompting
5:  \\ *Policy agent sampling*
6:  **for all** $i \in \{1, \ldots, N_a\}$ **in parallel do**
7:      $u_i \sim \text{Unif}(0,1)$
8:      **if** $u_i < \varepsilon(\ell)$ **then**
9:          $\phi^{(i)} \sim \text{Uniform}(\Phi_{\text{p}})$
10:     **else**
11:         $\phi^{(i)} \leftarrow \varnothing$
12:     **end if**
13:     $a_t^{(i)} = (S^{(i)}, \kappa^{(i)}) \sim \boldsymbol{\pi_{\mathbf{p}}}(\cdot \mid s_t, \rho_p(t), \text{phrase} = \phi^{(i)})$
14:     $\text{PM}^{(i)} \leftarrow \{(d_k, P(h_k; \mathcal{D})) \mid v_k \in S^{(i)}\}$        ▷ Parent metadata for action $a_t^{(i)}$
15: **end for**
16: \\ *World model rollouts and evaluation*
17: **for all** $i \in \{1, \ldots, N_a\}$ **in parallel do**
18:     **for all** $j \in \{1, \ldots, N_w\}$ **in parallel do**
19:         $\beta_{i,j} \sim \text{Unif}(0,1)$
20:         **if** $\beta_{i,j} < \varepsilon(\ell)$ **then**
21:             $\psi^{(i,j)} \sim \text{Uniform}(\Phi_{\text{wm}})$
22:         **else**
23:             $\psi^{(i,j)} \leftarrow \varnothing$
24:         **end if**
25:         $(\hat{h}^{(i,j)}, \hat{d}^{(i,j)}) \sim \boldsymbol{\pi_{\mathbf{wm}}}\left(\cdot \;\middle|\; \{(h_k, d_k)\}_{v_k \in S^{(i)}}, \kappa^{(i)}, \rho_{wm}(t), \text{phrase} = \psi^{(i,j)}\right)$
26:         Evaluate $P(\hat{h}^{(i,j)}; \mathcal{D})$        ▷ Cache $P(\hat{h}^{(i,j)}; \mathcal{D})$ for selection/critics
27:     **end for**
28: **end for**
29: $\ell \leftarrow \ell + N_a N_w$
30: \\ *Select best rollout and update graph*
31: $(i_\star, j_\star) \leftarrow \arg\max_{i,j} P(\hat{h}^{(i,j)}; \mathcal{D})$
32: $v_\star \leftarrow \left(\hat{h}^{(i_\star, j_\star)}, \kappa^{(i_\star)}, \hat{d}^{(i_\star, j_\star)}, P(\hat{h}^{(i_\star, j_\star)}; \mathcal{D}), \text{PM}^{(i_\star)}\right)$        ▷ Create entailed node
33: $V_{t+1} \leftarrow V_t \cup \{v_\star\}; \quad E_{t+1} \leftarrow E_t \cup \{S^{(i_\star)} \xrightarrow{\kappa^{(i_\star)}} v_\star\}$        ▷ Insert entailed node and edge
34: $V_{t+1}^{\text{disc}} \leftarrow V_t^{\text{disc}} \cup \{(\hat{h}^{(i,j)}, \kappa^{(i)}, \hat{d}^{(i,j)}, P(\hat{h}^{(i,j)}; \mathcal{D}), \text{PM}^{(i)}) \mid (i,j) \neq (i_\star, j_\star)\}$        ▷ Accumulate non-entailed rollouts
35: **if** $P(\hat{h}^{(i_\star, j_\star)}; \mathcal{D}) > P(h^\star; \mathcal{D})$ **then**
36:     $v^\star \leftarrow v_\star$        ▷ Update best overall node (implicitly updates best heuristic)
37: **end if**
38: $s_{t+1} \leftarrow (s_t \cup \{v_\star\}) \setminus (S^{(i_\star)} \setminus \{v^\star\})$        ▷ Keep new node and best node; prune other used parents
39: \\ *Critic reflections*
40: **for** $i = 1$ **to** $N_a$ **do**
41:     $R_p(a_t^{(i)}) \leftarrow \frac{1}{N_w} \sum_{j=1}^{N_w} P(\hat{h}^{(i,j)}; \mathcal{D})$        ▷ Reward signal used to rank actions for the policy critic
42:     $\mathcal{B}_p^{(i)} \leftarrow \{(\hat{d}^{(i,j)}, P(\hat{h}^{(i,j)}; \mathcal{D}))\}_{j=1}^{N_w}$
43: **end for**
44: $\rho_p(t+1) \leftarrow \boldsymbol{\pi_{\mathbf{p\_critic}}}\left(\{(a_t^{(i)}, R_p(a_t^{(i)}), \mathcal{B}_p^{(i)})\}_{i=1}^{N_a}, s_t\right)$        ▷ $R_p$ used for action ranking; $\mathcal{B}_p^{(i)}$ summarizes rollout variation
45: $(i_{\min}, j_{\min}) \leftarrow \arg\min_{i,j} P(\hat{h}^{(i,j)}; \mathcal{D})$
46: $\mathbf{best} \leftarrow (\hat{h}^{(i_\star, j_\star)}, \hat{d}^{(i_\star, j_\star)}, P(\hat{h}^{(i_\star, j_\star)}; \mathcal{D}))$
47: $\mathbf{worst} \leftarrow (\hat{h}^{(i_{\min}, j_{\min})}, \hat{d}^{(i_{\min}, j_{\min})}, P(\hat{h}^{(i_{\min}, j_{\min})}; \mathcal{D}))$
48: $\rho_{wm}(t+1) \leftarrow \boldsymbol{\pi_{\mathbf{wm\_critic}}}(\mathbf{best}, \mathbf{worst})$
49: **return** $(V_{t+1}, E_{t+1}, s_{t+1}, V_{t+1}^{\text{disc}}, \rho_p(t+1), \rho_{wm}(t+1), v^\star, \ell)$

---

## F.4. Ablation on Hyperparameters of PathWise

In Section 4.2, we presented an ablation study using the step-by-step construction framework on TSP training and validation sets to evaluate performance across in-domain and out-of-distribution settings, focusing on the effectiveness of the core

architectural choices and high-level mechanisms of PathWise. In this section, we further analyze the sensitivity of PathWise to additional hyperparameters that control the breadth and depth of the entailment and search process. These ablations examine the robustness of PathWise to reasonable parameter variations and clarify the rationale behind the selected default settings in terms of performance stability and computational efficiency.

*Table 14.* Ablations on the number of policy actions per entailment step $N_a$ and the number of world model rollouts per action $N_w$ in PathWise. We report optimality gaps (%) on training (*TSP50*) and validation sets (*TSP20*, *TSP50*) for the step-by-step construction framework on TSP.

| Methods | TSP20 | TSP50 | TSP50 |
|---|---|---|---|
| PathWise ($N_a$=2, $N_w$=2) | 6.08% | 9.72% | 8.79% |
| $N_a$=2, $N_w$=1 | 7.70% | 10.76% | 9.64% |
| $N_a$=2, $N_w$=3 | 7.16% | 10.36% | 9.38% |
| $N_a$=3, $N_w$=1 | 7.35% | 11.39% | 10.10% |
| $N_a$=3, $N_w$=2 | 6.67% | 10.06% | 9.16% |
| $N_a$=3, $N_w$=3 | 9.03% | 11.44% | 10.70% |

**Ablation on the number of policy actions and world model rollouts.**  Table 14 studies the effect of the number of policy actions per entailment step $N_a$ and the number of world model rollouts per action $N_w$. These parameters control the local branching factor and the amount of simulated feedback available to the policy and critic agents. The results show that configurations with $N_a$>1 and $N_w$>1 achieve comparable performance, indicating that both multiple actions and multiple rollouts are required for effective entailment.

Comparisons between increasing $N_a$ and increasing $N_w$ show that generating more candidate actions is more beneficial than adding additional rollouts for the same action. For example, increasing $N_a$ with moderate $N_w$ ($N_a$=3, $N_w$=2) outperforms increasing $N_w$ with fewer actions ($N_a$=2, $N_w$=3). This is consistent with the design of the policy critic, which operates on compact heuristic descriptions rather than full executable code, and aligns with the critic ablations in Table 3 that highlight the dominant role of policy-level feedback. Exposing the critic to a more diverse set of action candidates therefore provides stronger comparative signals than repeatedly simulating the same action through additional world model rollouts.

Increasing both $N_a$ and $N_w$ to larger values ($N_a$=3, $N_w$=3) degrades performance due to the excessive amount of context passed to the policy and critic agents. Conversely, reducing the configuration to smaller values, such as ($N_a$=2, $N_w$=1), limits rollout diversity and reduces the comparative signals available to the critics. Overall, these observations support the default setting ($N_a$=2, $N_w$=2) as a stable and efficient choice.

*Table 15.* Ablations on the population size $N_p$ in PathWise. We report optimality gaps (%) on training (*TSP50*) and validation sets (*TSP20*, *TSP50*) for the step-by-step construction framework on TSP.

| Methods | TSP20 | TSP50 | TSP50 |
|---|---|---|---|
| PathWise ($N_p$=6) | 6.08% | 9.72% | 8.79% |
| $N_p$=7 | 6.32% | 10.12% | 9.16% |
| $N_p$=8 | 6.35% | 10.15% | 9.38% |
| $N_p$=10 | 6.59% | 11.28% | 10.11% |

**Ablation on population size $N_p$.**  Table 15 shows the effect of the population size $N_p$, which controls the number of heuristics retained across outer iterations. PathWise is insensitive to moderate increases in $N_p$, with similar performance for $N_p$=6, 7, 8. In contrast, increasing the population to $N_p$=10 degrades performance, as the larger context passed to the policy and policy critic during early entailment steps reduces their ability to reason about high-quality entailments. These results support the default choice $N_p$=6 as a balanced setting, providing a better performance–cost trade-off.

*Table 16.* Ablations on the maximum number of inner entailment steps $I_{\max}$ in PathWise. We report optimality gaps (%) on training (*TSP50*) and validation sets (*TSP20*, *TSP50*) for the step-by-step construction framework on TSP.

| Methods | TSP20 | TSP50 | TSP50 |
|---|---|---|---|
| PathWise ($I_{\max}$=3) | 6.08% | 9.72% | 8.79% |
| $I_{\max}$=4 | 6.24% | 10.20% | 9.12% |

**Ablation on inner entailment steps** $I_{\max}$. Table 16 evaluates the impact of the maximum number of inner entailment steps $I_{\max}$, which controls the depth of multi-step entailment within each outer iteration. The results show that PathWise is not sensitive to this parameter within the tested range, with no significant performance differences observed. This reflects a trade-off in which too few entailment steps limit multi-step reasoning, while too many steps progressively reduce the available parent set and thereby reduce diversity, supporting the default choice $I_{\max}=3$.

In summary, these ablation results demonstrate that PathWise is not overly sensitive to its secondary hyperparameters within reasonable ranges, and they further justify the default parameter choices used in the main experiments.

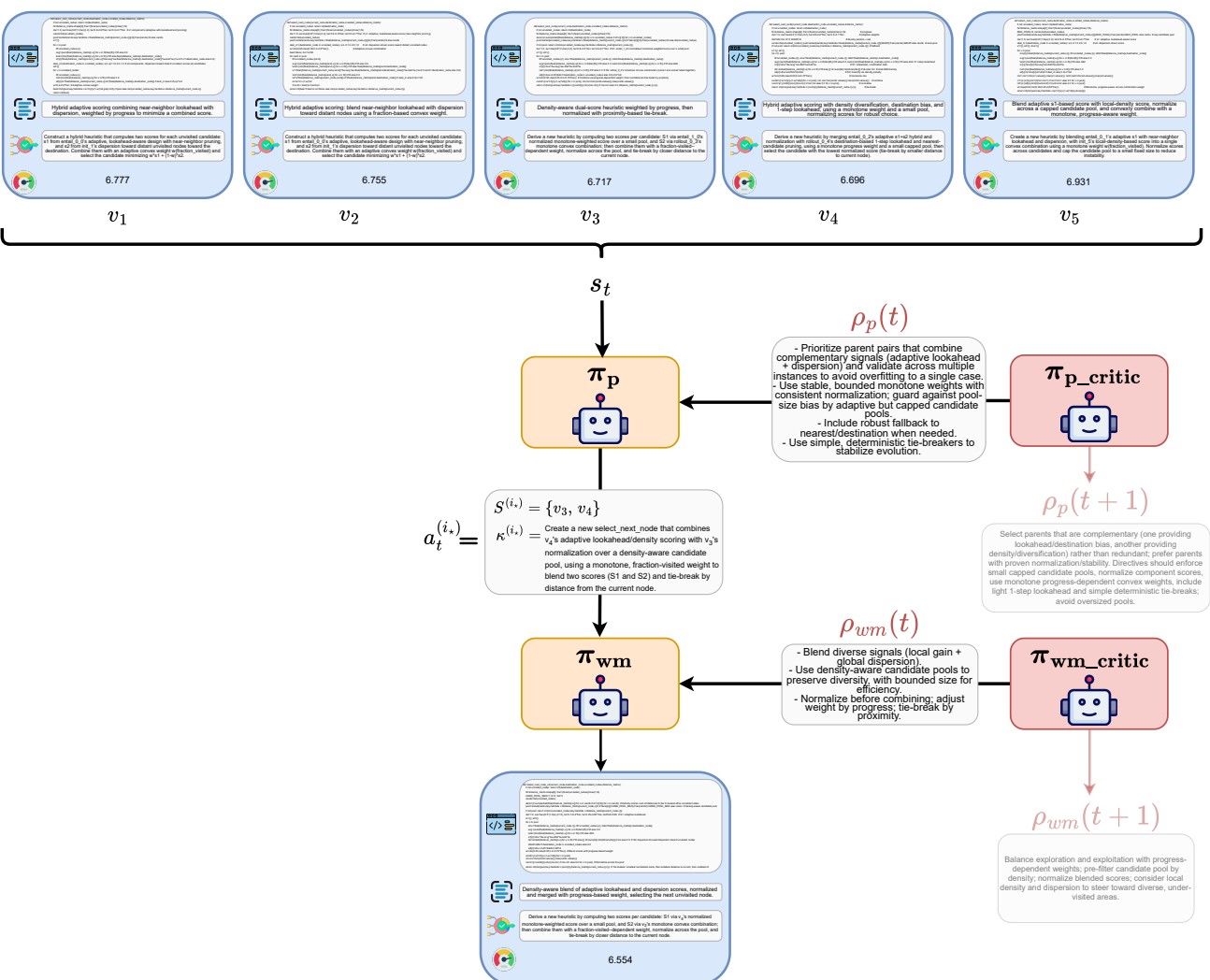

Figure 4. Example of PathWise at outer iteration $r = 1$ and entailment step $t = 2$, showing the entailed node $v_\star$. The current state $s_t$ contains nodes $\{v_1, \ldots, v_5\}$. The policy agent $\pi_p$ selects a parent set and generates a derivation rationale, which is executed by the world model $\pi_{wm}$ to entail a new node. The resulting entailed node is $v_\star$ with $i_\star = 0$ and $j_\star = 1$, as shown. The policy and world model critics provide routed reflections at step $t$, while shaded boxes indicate the updated reflections at step $t + 1$, conditioning the next step.

## G. Output Examples of PathWise

In this section, we provide example outputs of PathWise, where Figures 4 and 5 visualize representative entailment steps during heuristic design for TSP using `GPT-5-nano (medium)` within the constructive framework. Each figure highlights the entailed node selected at a given outer iteration and inner entailment step, together with the associated policy action, world model rollout, and routed critic feedbacks that guide the entailment of $v_\star$. Due to space constraints, the heuristic code displayed in the nodes is simplified and refactored by `GPT-5.2`, and parent metadata is omitted. In the derivation

rationales of nodes in $s_t$, nodes are referred to by unique identifiers of the form `entail_X_Y` or `rollout_X_Y`, indicating whether the node is entailed or discarded at outer iteration $X$ and inner entailment step $Y$; otherwise, we use the notation $v$ for notational convenience in the figures.

We observe that parent selection by the policy agent is not driven purely by performance ranking; instead, it reflects higher-level reasoning over derivation history, diversity, and prior entailment context encoded in the graph state. In addition, feedback from the world model critic frequently emphasizes reducing algorithmic complexity and improving execution efficiency through code-level refinements, such as removing redundant computations. Finally, the heuristics often maintain an internal state—implemented via function attributes such as a step counter and progress-dependent variables—which allows the selection behavior to adapt over construction steps rather than remaining static throughout the solution process.

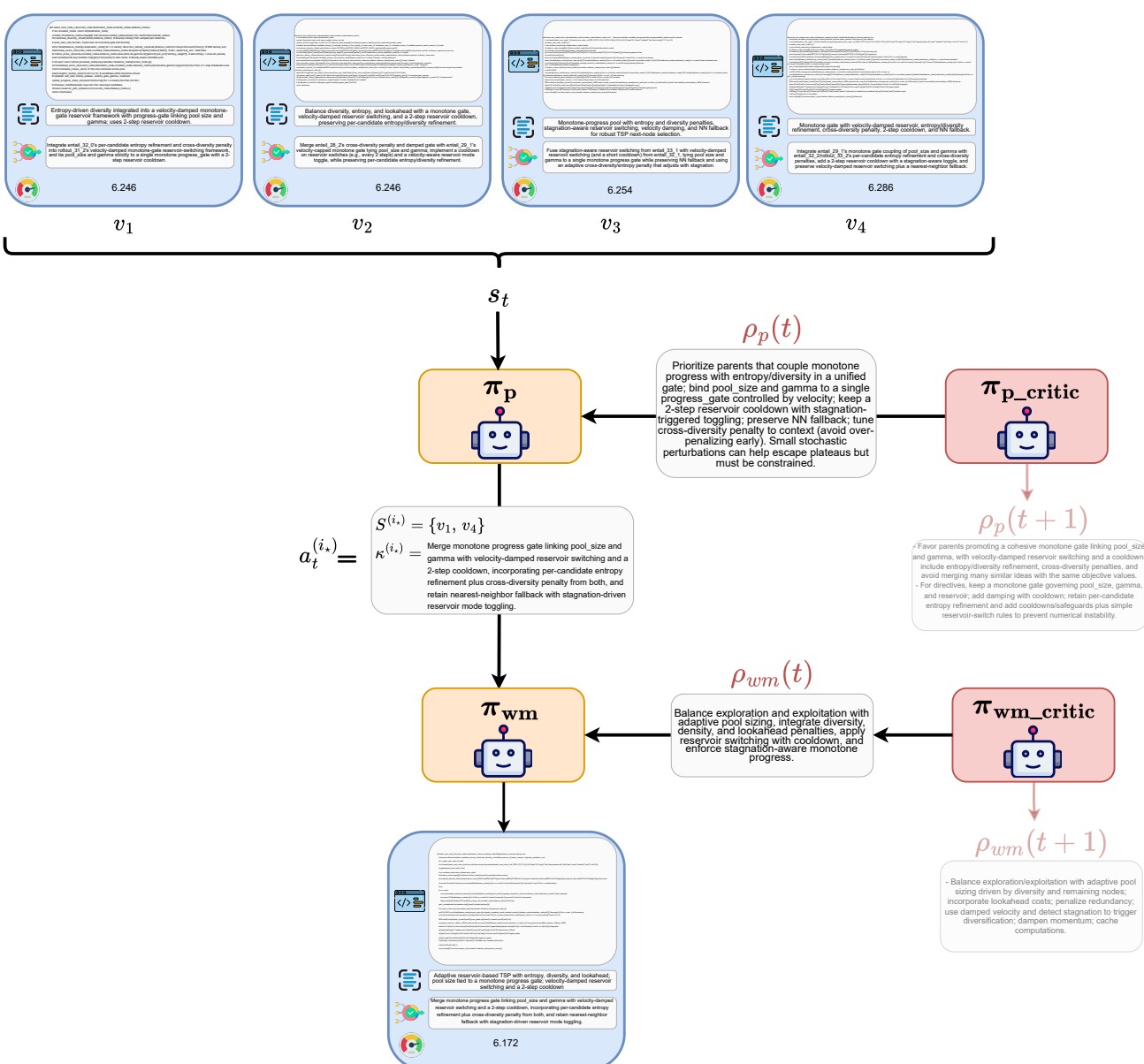

*Figure 5.* Example of PathWise at outer iteration $r = 34$ and entailment step $t = 1$, showing the entailed node $v_\star$. The current state $s_t$ contains nodes $\{v_1, \ldots, v_4\}$. The policy agent $\pi_p$ selects a parent set and generates a derivation rationale, which is executed by the world model $\pi_{wm}$ to entail a new node. The resulting entailed node is $v_\star$ with $i_\star = 1$ and $j_\star = 1$, as shown. The policy and world model critics provide routed reflections at step $t$, while shaded boxes indicate the updated reflections at step $t + 1$, conditioning the next step.

*Table 17.* Training time, token usage, and estimated monetary cost of LLM-based AHD methods across different LLMs.

| Task | Method | Training Time (hrs) | Input Tokens | Output Tokens | Overall Token Cost ($) |
|---|---|---|---|---|---|
| | | LLM-based AHD: *GPT-4o-mini* | | | |
| TSP-Constructive | ReEvo | 0.34 | 814,870 | 276,557 | 0.29 |
| | HSEvo | 0.56 | 516,752 | 153,691 | 0.17 |
| | MCTS-AHD | 2.10 | 690,216 | 228,349 | 0.24 |
| | PathWise (Ours) | 0.89 | 2,093,699 | 281,979 | 0.48 |
| MKP-ACO | ReEvo | 1.12 | 1,027,245 | 340,184 | 0.36 |
| | HSEvo | 1.00 | 448,491 | 129,941 | 0.15 |
| | MCTS-AHD | 3.10 | 845,442 | 266,019 | 0.29 |
| | PathWise (Ours) | 1.12 | 2,000,540 | 235,161 | 0.44 |
| Offline BPP-ACO | ReEvo | 1.73 | 849,334 | 281,124 | 0.30 |
| | HSEvo | 1.52 | 441,953 | 126,667 | 0.14 |
| | MCTS-AHD | 3.50 | 695,488 | 206,948 | 0.23 |
| | PathWise (Ours) | 1.72 | 2,125,645 | 252,723 | 0.47 |
| | | LLM-based AHD: *GPT-5-nano* (reasoning: low) | | | |
| TSP-Constructive | ReEvo | 0.37 | 1,199,659 | 678,508 | 0.33 |
| | HSEvo | 0.66 | 1,452,251 | 755,014 | 0.37 |
| | MCTS-AHD | 3.00 | 1,541,230 | 868,033 | 0.42 |
| | PathWise (Ours) | 1.19 | 3,583,414 | 862,143 | 0.52 |
| MKP-ACO | ReEvo | 0.69 | 2,087,945 | 851,566 | 0.45 |
| | HSEvo | 2.55 | 1,410,072 | 737,132 | 0.37 |
| | MCTS-AHD | 2.70 | 1,526,566 | 792,639 | 0.39 |
| | PathWise (Ours) | 1.28 | 4,841,499 | 846,706 | 0.58 |
| Offline BPP-ACO | ReEvo | 0.62 | 1,611,296 | 637,464 | 0.34 |
| | HSEvo | 1.74 | 1,064,308 | 505,791 | 0.26 |
| | MCTS-AHD | 3.30 | 1,320,616 | 800,247 | 0.39 |
| | PathWise (Ours) | 1.34 | 3,053,028 | 801,179 | 0.47 |
| | | LLM-based AHD: *GPT-5-nano* (reasoning: medium) | | | |
| TSP-Constructive | ReEvo | 2.05 | 2,377,468 | 3,226,613 | 1.41 |
| | HSEvo | 1.53 | 1,811,748 | 2,763,571 | 1.20 |
| | MCTS-AHD | 5.90 | 1,232,508 | 2,887,261 | 1.22 |
| | PathWise (Ours) | 5.05 | 6,360,001 | 2,901,738 | 1.48 |
| MKP-ACO | ReEvo | 1.32 | 1,888,283 | 2,269,843 | 1.00 |
| | HSEvo | 2.99 | 2,010,936 | 2,724,171 | 1.19 |
| | MCTS-AHD | 6.40 | 1,172,054 | 2,673,922 | 1.13 |
| | PathWise (Ours) | 5.20 | 5,328,958 | 2,814,809 | 1.39 |
| Offline BPP-ACO | ReEvo | 1.49 | 1,617,676 | 2,016,702 | 0.89 |
| | HSEvo | 2.78 | 1,748,237 | 2,685,562 | 1.16 |
| | MCTS-AHD | 7.32 | 1,692,991 | 3,039,435 | 1.30 |
| | PathWise (Ours) | 5.02 | 5,959,261 | 3,051,760 | 1.52 |

# H. Cost Analysis

This section reports the computational and monetary costs associated with LLM-based AHD methods. We decompose cost into two primary components: (i) wall-clock training time, and (ii) LLM token usage (input and output) together with the resulting monetary cost induced by model-specific pricing. These metrics provide a complementary perspective to solution quality by characterizing the practical efficiency and scalability of each method.

**Runtime Considerations.** Training time in LLM-based AHD is influenced not only by the number of heuristic evaluations but also by the computational complexity of the generated heuristics themselves. Heuristics that contain expensive operations, nested loops, or inefficient data structures can significantly increase per-evaluation runtime, even when the evaluation budget is fixed. As a result, methods that generate structurally complex or poorly optimized heuristics may incur higher wall-clock costs despite using the same number of evaluations. Conversely, generating simpler and more execution-efficient heuristics can substantially reduce overall training time.

**Token Usage & Model Pricing.** In LLM-based AHD methods, LLM-related cost is driven by both input tokens (prompt context, parent heuristics, metadata, and reflections) and output tokens (generated heuristic code, descriptions, and reasoning traces). In this paper, GPT-4o-mini and GPT-5-nano are used as the underlying LLMs for heuristic generation. Their pricing follows the official OpenAI specifications[2]. For both models, output tokens are substantially more expensive than input tokens. Specifically, for both GPT-4o-mini and GPT-5-nano, output tokens are approximately $8\times$ more expensive than input tokens. As a consequence, output token generation is the dominant contributor to total monetary cost, making methods that reduce unnecessary generations or verbose outputs significantly more cost-efficient.

**Analysis.** Table 17 summarizes training time, token usage, and total cost for different AHD methods across tasks. Across tasks and LLM configurations, PathWise achieves training times that are comparable to ReEvo (Ye et al., 2024) and

---

[2]https://platform.openai.com/docs/pricing

HSEvo (Dat et al., 2025), while reducing wall-clock training time by half relative to MCTS-AHD (Zheng et al., 2025). While ReEvo and HSEvo exhibit comparable training times, the heuristics they generate are consistently weaker in solution quality, as reflected in Table 1, Table 2, and Appendix E. Notably, despite having comparable output token usage, MCTS-AHD exhibits substantially longer training time, indicating that for LLM-based AHD the runtime overhead mainly arises from the inefficiency of the generated heuristics and their evaluation cost rather than from LLM API latency or the autoregressive nature of output token generation.

In terms of token usage, PathWise uses more input tokens than prior methods. This increase is expected, as parent selection and directive generation are explicitly handled by the LLM, requiring richer contextual inputs such as parent metadata and heuristic representations. At the same time, PathWise avoids providing full heuristic code whenever possible and instead relies on compact summaries and structured metadata to reduce unnecessary input length. Importantly, this additional input does not lead to prohibitive cost, since output tokens remain the dominant cost factor, allowing PathWise to maintain competitive monetary cost while achieving stronger performance.

Across all methods, increasing the reasoning level of GPT-5-nano leads to higher output token generation due to the generation of internal reasoning tokens, as shown in Table 17. Higher levels of explicit reasoning are known to produce longer outputs and higher token consumption in LLMs (Wei et al., 2022; Zhou et al., 2023a; Ma et al., 2025), highlighting the importance of controlling reasoning verbosity when scaling LLM-based AHD methods.

In many practical applications, training time is often a more critical constraint than monetary cost. As training is extended to NP-hard problems that require learning heuristics on larger instances or involve expensive heuristic evaluations (e.g., large-scale TSP or CVRP instances, high-dimensional packing and scheduling problems), or to combinatorial problems coupled with costly simulation or experimentation, methods with long training cycles become impractical due to the cumulative overhead of repeated heuristic evaluations. Such expensive evaluations commonly arise in scientific and engineering settings, including adaptive experimental design for combinatorial structures (Doppa, 2021), protein and molecular sequence optimization (Yuan et al., 2022; Qiu et al., 2024; Reinhart & Statt, 2024), materials discovery with sequential experimentation (Qian et al., 2023; Chitturi et al., 2024), and high-dimensional black-box optimization over combinatorial or mixed spaces (Papenmeier et al., 2023).

## I. Extended Discussions

This section discusses limitations of PathWise and LLM-based AHD methods in general, followed by directions for future work.

### I.1. Limitations

PathWise enhances AHD by combining a hybrid graph-based and population-based formulation with state-aware planning through reasoning over an entailment graph. This structured representation enables memory of derivation history and more informed evolutionary decisions. However, several limitations remain. First, heuristic generation relies on stochastic LLM outputs, and even with structured planning and critic feedback, generated implementations may vary in quality, leading to temporary instability. Such variability can reduce effective diversity in the entailment graph, limiting the contrast available to critic agents and weakening feedback at some steps. More broadly, this behavior is common to LLM-based AHD methods, as they rely on black-box LLMs whose stochastic sampling introduces non-determinism into both heuristic generation and the overall search process, causing runs to diverge even under fixed settings. Second, heuristic performance varies across problem domains when different LLM backbones or reasoning levels are used. This variation likely arises from differences in model training and post-processing (Slocum et al., 2025; Sun et al., 2025; Lanchantin et al., 2025), which can directly impact the diversity of generated heuristics. In addition, different backbones tend to produce heuristics with distinct implementation styles, such as differences in code complexity or abstraction level, and the impact of these differences can vary across problem domains.

Due to the multi-agent design of PathWise and the inherent stochasticity of LLM-based generation, both intermediate trajectories and final outcomes can vary across backbones and problem settings. In our experiments, this variability was most pronounced in the early stages of the search, where some runs exhibited limited early progress and met the stopping criterion before reaching the more stable improvement patterns observed in other runs. This suggests that PathWise is sensitive to early-step randomness, and improving robustness across runs remains an important direction for future work.

### I.2. Future Work

Future work includes extending PathWise to settings where heuristic evaluation is significantly more expensive and training time becomes a primary constraint. In many real-world NP-hard optimization problems, learning effective heuristics requires operating on large instances or under costly evaluation pipelines coupled with simulation or experimentation. In such problems, long evolutionary cycles with repeated evaluations are often impractical. PathWise offers a natural direction by allowing policy, world model, and critic agents to be trained to capture problem-domain knowledge and reusable evolutionary strategies in a state-aware manner, rather than relying on fixed search parameters or static operators. Developing training algorithms that balance efficiency with the diversity required for effective heuristic discovery is a promising direction.

## J. License

The licenses and URLs of baseline methods and software resources are provided in Table 18.

*Table 18.* A summary of licenses.

| Resources | Type | License | URL |
|---|---|---|---|
| LKH3 | Code | Available for academic research use | http://webhotel4.ruc.dk/~keld/research/LKH-3/ |
| OR-Tools | Code | MIT License | https://developers.google.com/optimization/pack/knapsack?hl=zh-cn |
| POMO | Code | Available online | https://github.com/yd-kwon/POMO/tree/master |
| ACO/DeepACO | Code | MIT License | https://github.com/henry-yeh/DeepACO |
| VRP-DACT | Code | MIT License | https://github.com/yining043/VRP-DACT |
| NeuOpt | Code | MIT License | https://github.com/yining043/NeuOpt |
| Funsearch | Code | Apache License | https://github.com/google-deepmind/funsearch |
| EoH | Code | MIT License | https://github.com/FeiLiu36/EoH/tree/main |
| ReEvo | Code | MIT License | https://github.com/ai4co/reevo |
| HSEvo | Code | Available online | https://github.com/datphamvn/HSEvo |
| MCTS-AHD | Code | Available online | https://github.com/zz1358m/MCTS-AHD-master |

