# OpenReview forum: "PathWise: Planning through World Model for Automated Heuristic Design via Self-Evolving LLMs"
_ICML.cc/2026/Conference — ICML 2026 regular_

### Official Review · Reviewer_1rPG · 2026-02-27

**Soundness:** 3
**Presentation:** 3
**Significance:** 3
**Originality:** 3
**Overall Recommendation:** 5
**Confidence:** 5

**Summary:**

This paper proposes PathWise, a budgeted search framework for automatic heuristic design (AHD). It models heuristic evolution as an MDP and maintains an entailment graph as compact search memory. At each step, a policy agent selects parent heuristics and a natural-language rationale; a world-model agent generates multiple candidate heuristics (rollouts), evaluates them, and adds the best to the graph; critic agents produce reusable reflections to guide future steps. An outer-loop population update (leaf-first) plus exploration tricks (prompt perturbation, state shuffling) improves diversity and robustness.

This paper is well-written, with clear demonstrations of its effectiveness. I believe Pairwise can become a newer and better framework.

**Compliance With Llm Reviewing Policy:**

Affirmed.

**Final Justification:**

This paper proposes PathWise, a budgeted search framework for automatic heuristic design (AHD). It models heuristic evolution as an MDP and maintains an entailment graph as compact search memory. This paper is well-written and quite novel, I will suport its acceptance.

**Key Questions For Authors:**

1. This paper follows the rigorous writing style of MCTS-AHD, but I suggest referring to MCTS-AHD and incorporating a discussion on significance (possibly p-test).
2. Minor Typos :
* Line 275: step-by-step heuristic construction
* Line276: OR-Tools citation no year.

**Limitations:**

well discussed

**Strengths And Weaknesses:**

**Strengths:**


I recommend this paper to be accepted.

1. This article is quite clear, and the reasoning regarding entailment aligns with intuition.

2. It discusses many details, especially the overhead of evaluation and LLM calls, which is very well written.

3. There are clear improvements in the effectiveness.

**Weaknesses:**

No major weaknesses, please refer to the questions for minor problems.

---

> ### Author Rebuttal · Authors · 2026-03-30
>
> We sincerely thank the reviewer for the detailed and encouraging evaluation. We are glad that you find PathWise to be well-written with clear demonstrations of its effectiveness, and that the entailment-based reasoning aligns with intuition. We address the suggestions below.
>
> ---
>
> ## Suggestion 1: Statistical Testing
> We conducted 6 independent runs of PathWise and baselines across multiple problems and LLM backbones. We report mean gap, standard deviation (std), and p-values from single-tailed t-tests between PathWise and each baseline on each test set size.
>
> **Table R1. Statistical significance of PathWise on TSP-ACO. Gap (%), std, and p-value on test sets (N=50, N=100).**
>
> GPT-5-nano (medium):
> | Method | Gap↓ (N=50) | Std | p-value | Gap↓ (N=100) | Std | p-value |
> |---|---|---|---|---|---|---|
> | ReEvo | 7.70% | 3.62% | 0.0047 | 14.19% | 4.79% | 0.0015 |
> | HSEvo | 4.77% | 0.69% | 0.0014 | 10.91% | 2.03% | 0.0006 |
> | PathWise | 1.93% | 1.41% | - | 4.67% | 2.62% | - |
>
> DeepSeek-V3.2:
> | Method | Gap↓ (N=50) | Std | p-value | Gap↓ (N=100) | Std | p-value |
> |---|---|---|---|---|---|---|
> | ReEvo | 3.53% | 0.98% | 0.0034 | 8.59% | 2.96% | 0.0043 |
> | HSEvo | 2.77% | 1.00% | 0.0403 | 6.40% | 2.74% | 0.0377 |
> | PathWise | 1.86% | 0.41% | - | 3.78% | 1.50% | - |
>
> Note: MCTS-AHD is excluded from TSP-ACO due to its long training time per run.
>
> **Table R2. Statistical significance of PathWise on KP-Constructive. Gap (%), std, and p-value on test sets (N=50 W=12.5, N=100 W=25).**
>
> GPT-4o-mini:
> | Method | Gap↓ (N=50) | Std | p-value | Gap↓ (N=100) | Std | p-value |
> |---|---|---|---|---|---|---|
> | MCTS-AHD | 0.2583% | 0.0133% | 0.0002 | 0.1117% | 0.0098% | 0.0096 |
> | PathWise | 0.2217% | 0.0075% | - | 0.0983% | 0.0041% | - |
>
> GPT-5-nano (medium):
> | Method | Gap↓ (N=50) | Std | p-value | Gap↓ (N=100) | Std | p-value |
> |---|---|---|---|---|---|---|
> | MCTS-AHD | 0.2167% | 0.0151% | 0.0020 | 0.0900% | 0.0126% | 0.0026 |
> | PathWise | 0.1117% | 0.0538% | - | 0.0517% | 0.0214% | - |
>
> Results confirm that PathWise improves over baselines (all p < 0.05) across problems, sizes, and LLM backbones. We will include this statistical analysis in our manuscript.
>
> ---
>
> ## Suggestion 2: Minor Typos
> Thank you for pointing these out. We will fix both in our manuscript.

---

> > ### Author Rebuttal · Reviewer_1rPG · 2026-04-02
> >
> > Good work and nice rebuttal, I will support.

---

> > > ### Author Response · Authors · 2026-04-06
> > >
> > > Thank you for your support of the paper. We greatly appreciate your constructive feedback.

---

### Official Review · Reviewer_upCx · 2026-03-09

**Soundness:** 3
**Presentation:** 3
**Significance:** 3
**Originality:** 3
**Overall Recommendation:** 4
**Confidence:** 3

**Summary:**

This paper proposes PathWise, a multi-agent framework for automated heuristic design in combinatorial optimization. It models heuristic discovery as sequential decision-making over an entailment graph that records heuristic derivations and performance. A policy agent selects heuristic transformations, a world model agent generates new heuristic code, and critic agents provide feedback to guide future steps. Experiments on several optimization tasks show faster convergence and stronger performance to prior LLM-based AHD methods.

**Compliance With Llm Reviewing Policy:**

Affirmed.

**Ethical Review Concerns:**

1. The paper formulates the problem as an MDP, but it does not appear to optimize the policy in the RL sense. The method seems training-free and relies mainly on prompting and search. Could the authors provide more intuition on why this approach is expected to work?

2. The performance of the framework appears to depend heavily on the capability of the underlying base model. Do the authors expect the framework to generalize to problem scales that a single base model cannot handle?

**Final Justification:**

I will keep my score as a weak accept. While the authors have addressed most of the concerns, I still feel that describing “natural-language reflections” as a reinforcement learning problem may be misleading, as it could suggest that the LLM policy is being directly optimized. Nevertheless, the experimental results are strong, so I keep weak accept.

**Key Questions For Authors:**

1. The paper formulates the problem as an MDP, but it does not actually optimize the policy in the RL sense. I feel it is a training-free method. Could you explain it?

2. I feel the performance of this framework is largely limited by the base model. Do you think this framework will generalize to the size of problem that the single base model can not solve?

**Limitations:**

Yes

**Strengths And Weaknesses:**

Strengths
1. The interaction among the policy agent, world model agent, and critic agents clearly separates planning, execution, and evaluation.

2. The experimental evaluation is comprehensive, including multiple baselines with different LLM backbones and problem scales.

3. The ablation studies are thorough and provide clear evidence of the contribution of each component.

Weaknesses:
1. The framework combines several existing ideas (world models, multi-agent reasoning, reflection, evolutionary search), and the main novelty appears to be applying LLM-based memory and planning to combinatorial optimization.

2. The writing is not sufficiently clear. The overall framework and method are difficult to grasp from a single reading of the method section.

---

> ### Author Rebuttal · Authors · 2026-03-30
>
> We thank the reviewer for the constructive review and for recognizing the clear separation of planning, execution, and evaluation in our multi-agent design. We first address the questions, followed by weaknesses.
>
> ## Question 1: MDP Formulation and Training-Free Nature
> PathWise is indeed training-free in the sense that no model parameters are updated. However, the MDP formulation serves as the structural backbone that defines states (entailment graph frontier), actions (parent selection + derivation rationale), state transitions (updated entailment graph), and rewards (derived from heuristic performance). This structure enables systematic, state-aware planning through the world model rather than ad-hoc prompting.
>
> Instead of optimizing the policy through gradient-based updates, PathWise reinforces the policy and world model through natural-language reflections from critic agents [1, 2, 3], shifting their effective behavior at each step without changing language model parameters. In our framework, the policy critic compares actions and their resulting rollouts to provide strategic feedback, while the world model critic contrasts best and worst rollouts to guide code synthesis. The detailed procedure is provided in Algorithm 3 (Appendix F.3).
>
> This training-free design is a deliberate feature: it makes PathWise problem-agnostic (no need to train separately for each COP), applicable with closed-source LLMs where parameter updates are not possible, and directly applicable across different search frameworks without retraining. Both the world model and the policy agents rely on the internal knowledge of the LLM for predicting the next state and drawing actions, respectively.
>
> ## Question 2: Dependence on Base Model and Scalability
> It is important to note that in LLM-based AHD, the base model does not solve problem instances directly. Instead, the LLM generates heuristic *code* that is then executed on instances of any size. The LLM never sees the actual problem instances during heuristic generation. Therefore, problem scale is not bounded by the LLM's context window or reasoning capacity.
>
> Our experiments across multiple COPs and scales support this. PathWise heuristics are designed on small training instances (e.g., TSP with N=50) and then evaluated on much larger test instances (e.g., N=100, 200). As reported in Tables 1-2, 7-9, PathWise exhibits strong out-of-distribution generalization, with performance improvements that grow as problem size increases. For example, on KP, the mean relative gap improvement (MRGI) increases from 31.67% on the in-domain test set (N=100, W=25) to 81.34% on the largest out-of-domain test set (N=500, W=25) when averaging across LLM backbones (Section 4.1). Similarly, on OP with ACO, the average MRGI increases from 25.40% (N=50) to 84.22% (N=200) (Section 4.1).
>
> Regarding base model capability, our experiments across three backbones (GPT-4o-mini, GPT-5-nano low/medium) in Tables 1-2, 7-9 show that PathWise achieves better results across different backbones. Please also see our response to `Reviewer rgzX` (Table R2) where PathWise generalizes to the open-source model DeepSeek-V3.2.
>
> ---
>
> ## Weakness 1: Writing Clarity
> We appreciate this feedback. We will improve the readability of the method section by adding a high-level overview paragraph at the beginning of Section 3 that summarizes the full pipeline before the technical details.
>
> ---
>
> ## References
>
> [1] N. Shinn, F. Cassano, A. Gopinath, K. R. Narasimhan, and S. Yao. "Reflexion: Language Agents with Verbal Reinforcement Learning." In *Proc. NeurIPS*, 2023.
>
> [2] A. Madaan, N. Tandon, P. Gupta, S. Hallinan, L. Gao, S. Wiegreffe, U. Alon, N. Dziri, S. Prabhumoye, Y. Yang, et al. "Self-Refine: Iterative Refinement with Self-Feedback." In *Proc. NeurIPS*, 2023.
>
> [3] Q. Zhang, C. Hu, S. Upasani, B. Ma, F. Hong, V. Kamanuru, J. Rainton, C. Wu, M. Ji, H. Li, et al. "Agentic Context Engineering: Evolving Contexts for Self-Improving Language Models." *arXiv:2510.04618*, 2025.

---

> > ### Author Rebuttal · Reviewer_upCx · 2026-04-01
> >
> > Thank you so much, all my questions are resolved.

---

> > > ### Author Response · Authors · 2026-04-06
> > >
> > > Thank you for acknowledging that our rebuttal fully resolved your questions, and for your positive evaluation. We hope our responses will be reflected positively in your final assessment of the paper.

---

### Official Review · Reviewer_xTWv · 2026-03-13

**Soundness:** 3
**Presentation:** 4
**Significance:** 4
**Originality:** 3
**Overall Recommendation:** 4
**Confidence:** 3

**Summary:**

The paper proposes PathWise, a multi-agent framework for automated heuristic design (AHD) that treats heuristic generation as a sequential decision process over an entailment graph serving as a compact, stateful memory of the search. A policy agent plans parent selection and natural-language derivation rationales, a world model agent synthesizes heuristic code conditioned on those actions, and critic agents produce routed reflections to adapt subsequent decisions; prompt-level diversity mechanisms further encourage exploration. Across multiple combinatorial optimization problems (e.g., TSP, KP, CVRP, MKP, OP, BPP) and search frameworks (constructive, ACO, GLS), PathWise reports faster convergence and better final heuristics than recent LLM-based AHD baselines.

**Compliance With Llm Reviewing Policy:**

Affirmed.

**Final Justification:**

The authors provided a comprehensive and convincing rebuttal that addressed all my primary concerns. Specifically:

Entailment Graph vs. Population (W1): The newly added ablations clearly isolate the contribution of the graph structure, proving that state pruning and multi-step chained derivation offer tangible benefits over matched population-only variants.

Critic Strategy (W3): The additional experiments successfully justify the "best-vs-worst" contrasting method as not only the most performant but also the most context-efficient strategy compared to full ranking or top-2 comparisons.

Robustness & Statistics (W2, W4): The inclusion of 6 independent runs with t-tests and the clarification on how per-step regeneration mitigates textual hallucination noise significantly improve the empirical soundness of the work.

**Key Questions For Authors:**

1. While the ablation studies demonstrate the value of the critic agents and prompt perturbation, there is no explicit ablation isolating the contribution of the entailment graph itself. How much of the performance gain is strictly due to the stateful graph memory versus the multi-agent reflection design? Could the authors provide an ablation using a matched, purely population-based variant (without the entailment graph) that utilizes the exact same policy, world model, and reflection agents?
2. The system relies entirely on textual reflections (verbal gradients) for policy and world model adaptation, rather than incorporating learned value models or RL-style credit assignment. How robust is this text-based adaptation to noisy, inconsistent, or hallucinated feedback from the critic LLMs? Have the authors analyzed the stability of the evolutionary trajectory over extended horizons when relying solely on these textual updates?
3. Currently, the world model critic generates reflections by contrasting only the best-performing and worst-performing heuristic rollouts. Did the authors experiment with richer supervision signals, such as providing the critic with a full ranking of all generated candidates or utilizing pairwise preference modeling across the batch? Why was the strictly best-vs-worst comparison chosen over more comprehensive ranking information?

**Limitations:**

yes

**Strengths And Weaknesses:**

## Strength
- Automated design of heuristics across diverse COPs is an important and timely challenge; enabling state-aware, semantic planning with memory addresses known limitations of LLM-AHD (myopia, redundancy, rediscovery).
- The approach is LLM-agnostic and appears cost-aware (small Na, Nw, Imax), making it practical for broader use.
- Casting AHD as planning over an entailment graph with stateful, semantically meaningful derivation history is a compelling departure from stateless population methods and purely performance-driven tree search.
- The overall system design is clearly depicted (two-timescale loop, entailment steps, critics); the MDP formulation, action structure, and graph update rule are explicit.

## Weakness
1. No explicit ablation isolating the entailment-graph/stateful planning benefit versus a matched, purely population-based variant using the same agents and reflections.
2. The policy/world model adaptation relies on textual reflections rather than learned value models or bandit/RL-style credit assignment; stability and robustness to noisy feedback are not deeply analyzed.
3. The world model critic compares only best vs. worst candidates; richer ranking or pairwise preference modeling might yield stronger supervision.
4. Statistical testing and more exhaustive reporting of variance (beyond a few plots) are limited; only three runs are used for most tabled results.

---

> ### Author Rebuttal · Authors · 2026-03-30
>
> We thank the reviewer for the thoughtful review and for recognizing the entailment graph formulation as a compelling departure from existing methods. We first address the questions, followed by weaknesses.
> ## Question 1: Ablation Isolating the Entailment Graph
> We conduct two without-entailment-graph ablations that isolate the contribution of the graph. Both variants are population-based with the same multi-agent architecture: they generate $N_a \times N_w \times I_{\max}$ heuristics per generation, apply elitist replacement, and maintain a fixed population of size $N_p$. Critic reflections are produced after each generation.
> - **Population + Historical Context**: Population-based variant that preserves historical context (a running log of previously generated heuristics with their derivation rationales and parent metadata shown to agents) in prompts, but without graph structure.
> - **Population-Only**: Purely population-based variant that excludes all historical context, derivation rationale, and parent metadata from agents.
>
> **Table R1. Ablations on the entailment graph in PathWise for TSP-Constructive. We report optimality gaps (%) on the validation set (TSP50).**
> | Method | TSP50 |
> |---|---|
> | PathWise | 9.72% |
> | Population + Historical Context | 14.04% |
> | Population-Only | 16.82% |
>
> The entailment graph provides a stateful representation of the search process, and its removal leads to clear degradation as shown in Table R1, attributed mainly to two mechanisms absent in both ablated variants. First, state pruning (Section 3.1) keeps the agent context compact at each step through downselection of parent nodes after entailment, whereas both population-based variants present all $N_p$ heuristics to agents every step, exposing them to a broad search space that degrades reasoning quality and reduces agent efficiency, especially in the policy and its critic. Second, the graph enables chained derivation across inner entailment steps, where each child builds on the modifications of its parents. Without the graph, all heuristics are derived directly from the same flat pool at depth 1, discarding intermediate heuristics and preventing the compounding refinement that multi-step entailment enables. Population-Only performs worst, confirming that both structured memory and historical awareness contribute to PathWise's performance, with the entailment graph serving as the effective medium through which the multi-agent architecture coordinates.
> ## Question 2: Robustness of Textual Reflections
> The evolution curves in Figures 1 and 3 show that PathWise achieves notably lower variance (shaded regions) than baselines across evolution steps, indicating consistent guidance rather than instability from noisy feedback. For further statistical evidence (p-values), please see our response to `Reviewer 1rPG`, where additional runs confirm improvements across problems and LLM backbones.
>
> While textual reflections can in principle be noisy, two design choices mitigate this: (1) reflections are regenerated at each step based on the latest outputs, so a single noisy reflection does not propagate; and (2) the best rollout selection (Section 3.2) acts as a performance-based filter, ensuring that the highest-quality heuristic is committed to the graph.
> ## Question 3: World Model Critic Comparison Strategy
> We conducted additional ablations comparing different world model critic strategies:
> - **Best-vs-Worst (default)**: Critic contrasts the best and worst rollouts.
> - **All Rollouts**: Critic receives all $N_a \times N_w$ rollouts.
> - **Top-2**: Critic contrasts the two best-performing rollouts.
>
> **Table R2. Ablations on the world model critic comparison strategy for TSP-Constructive. We report optimality gaps (%) on the validation set (TSP50).**
> | WM Critic Strategy | TSP50 |
> |---|---|
> | Best-vs-Worst (default) | 9.72% |
> | All Rollouts | 9.98% |
> | Top-2 | 10.36% |
>
> The best-vs-worst strategy achieves the strongest performance. All Rollouts performs comparably to the default but is less cost-efficient due to increased context length, and as shown in Table 11, excessive context degrades agent performance, suggesting this approach would scale poorly with higher $N_a$ and $N_w$. Top-2 performs worse because as heuristic evolution progresses, the performance variance among top rollouts decreases, reducing the contrastive signal available to the critic.
>
> ---
> We note that Weaknesses 1-3 are directly addressed by the questions above.
> ## Weakness 4: Statistical Testing and Variance
> We have conducted 6 independent runs across multiple problems (TSP-ACO, KP-Constructive) and LLM backbones (GPT-4o-mini, GPT-5-nano (medium), and DeepSeek-V3.2), reporting average gap, standard deviation, and p-values from single-tailed t-tests. Please see our response to `Reviewer 1rPG` (Tables R1-R2) for the full results. The analysis confirms statistically significant improvements of PathWise over baselines across settings.

---

> > ### Author Rebuttal · Reviewer_xTWv · 2026-04-03
> >
> > Thank you for the detailed clarification. My concerns have been largely addressed.

---

> > > ### Author Response · Authors · 2026-04-06
> > >
> > > Thank you for carefully considering our rebuttal and for updating your assessment. We greatly appreciate your constructive feedback.

---

### Official Review · Reviewer_rgzX · 2026-03-15

**Soundness:** 2
**Presentation:** 2
**Significance:** 2
**Originality:** 2
**Overall Recommendation:** 3
**Confidence:** 3

**Summary:**

This paper proposes PathWise (Planning through World Model for Automated Heuristic Design via Self-Evolving LLMs), a multi-agent reasoning framework that formulates LLM-based automated heuristic design (AHD) for combinatorial optimization problems (COPs) as a sequential decision process over an entailment graph. Existing LLM-based AHD methods, whether population-based (e.g., FunSearch, EoH, ReEvo, HSEvo) or tree-based (e.g., MCTS-AHD), suffer from myopic heuristic generation, redundant evaluations, and a lack of stateful memory about how heuristics are derived across generations. PathWise addresses these limitations through a hybrid graph-based and population-based architecture operating at two timescales: an outer loop maintains a population of root nodes, while an inner loop incrementally constructs an entailment graph that records derivation rationale, parent information, and performance history as a compact, stateful memory of the search trajectory. Three coordinated LLM agents drive the process: (1) a Policy Agent that selects parent heuristics and formulates natural-language derivation directives (replacing fixed evolutionary operators with semantic-level planning); (2) a World Model Agent that executes these actions by generating heuristic code rollouts; and (3) two Critic Agents (Policy Critic and World Model Critic) that provide routed reflections to guide subsequent steps. Additionally, PathWise introduces prompt-level diversity mechanisms,  diversity-aware prompt perturbation with decaying exploration rate and state shuffling to mitigate LLM positional bias, to ensure broader exploration. Extensive experiments across six COPs (TSP, CVRP, KP, MKP, OP, BPP) under three search frameworks (constructive, ACO, GLS) and three LLM backbones (GPT-4o-mini, GPT-5-nano low/medium) demonstrate that PathWise consistently discovers stronger heuristics with fewer evaluations (500 vs. 1000 for baselines), achieving mean relative gap improvements of 20–89% over prior LLM-based AHD methods depending on the problem, with faster convergence, lower variance, and strong out-of-distribution generalization to larger problem sizes.

**Compliance With Llm Reviewing Policy:**

Affirmed.

**Key Questions For Authors:**

1. All ablations (Tables 3–4) are conducted solely on TSP under the constructive framework with GPT-5-nano (low). Do the relative contributions of each component (e.g., Policy Critic being more important than World Model Critic) hold across other problems (e.g., CVRP-ACO) or other LLM backbones? What evidence do you have that these findings transfer?
2. The entailment graph stores derivation rationale, parent metadata, and performance history. Have you compared PathWise against a simpler baseline that simply includes a running log of past heuristics and their performance in the LLM prompt (i.e., richer context without the graph structure)? This would help isolate whether the improvement comes from the graph topology itself or just from providing more historical context to the LLM.
3. All experiments use proprietary OpenAI models (GPT-4o-mini, GPT-5-nano). Have you tested PathWise with open-source LLMs (e.g., Llama, DeepSeek-Coder)?
4. Given that LLM API outputs can vary across time (model updates, temperature stochasticity), how stable are the reported results if experiments are re-run months later? Have you observed significant variance across different time periods of API access?

**Limitations:**

Yes

**Strengths And Weaknesses:**

**Soundness:**

Strength:
1. Comprehensive experimental coverage. The evaluation spans six COPs (TSP, CVRP, KP, MKP, OP, BPP), three distinct search frameworks (constructive, ACO, GLS), and three LLM backbones (GPT-4o-mini, GPT-5-nano low/medium).
2. Thorough ablation studies. The paper systematically ablates each core component, both critic agents (Table 3), prompt perturbation, and state shuffling (Table 4)

Weakness:
1. All ablations (Tables 3–4) are conducted only on TSP under the constructive framework with a single LLM backbone (GPT-5-nano low). It is unclear whether the relative importance of each component (e.g., Policy Critic > World Model Critic) holds across other problems, search frameworks, or LLM backends.
2. Cost and efficiency analysis is relegated to the appendix. Since PathWise uses multiple LLM agents per step (policy + world model + 2 critics), the total number of LLM API calls per evaluation could be substantially higher than baselines that use a single LLM call per heuristic.

**Presentation:**

Strength:
1. The paper follows a clear logical progression. The paper is easy to follow in general.

Weakness:
1. Terms like "world model," "planning," and "MDP" carry strong connotations in the RL/planning literature, implying learned dynamics models and policy optimization. Here, the "world model" is simply an LLM prompted to generate code, and "planning" refers to structured prompting with derivation history. While the analogy is reasonable, the paper should more clearly acknowledge that these are metaphorical rather than operational uses of these concepts.

**Significance:**

Strength:
1. The framework is not tied to a specific problem or search paradigm. It works across constructive, ACO, and GLS frameworks, suggesting it could serve as a general-purpose improvement layer for LLM-based AHD.

Weakness:
1. Absolute quality gap to specialized solvers remains large. On TSP, even the best PathWise heuristic has an 8–13% gap to LKH-3 (Table 1). While this is expected for AHD, it limits the immediate practical impact for problems where strong specialized solvers exist.
2. Dependence on proprietary LLMs raises reproducibility concerns. All experiments use GPT-4o-mini and GPT-5-nano, which are closed-source commercial models. Model behavior can change across API versions, making exact reproduction difficult. The paper does not evaluate any open-source LLMs (e.g., Llama, DeepSeek), limiting the accessibility of the contribution.

**Originality:**

Strength:
1. The idea of maintaining a directed graph that records derivation rationale, parent metadata, and performance history as a compact search memory is, to the best of my knowledge, new in the AHD literature.

Weakness:
1. The policy–world model–critic decomposition closely mirrors standard LLM agent frameworks (e.g., planner + executor + verifier) that are widely used in the broader LLM agent literature. While the application to AHD is new, the architectural pattern itself is not novel.
2. The conceptual leap from MCTS-AHD's tree structure to PathWise's entailment graph is moderate, i.e., both maintain parent-child derivation relationships; PathWise adds richer node metadata and a different traversal/selection strategy.

---

> ### Author Rebuttal · Authors · 2026-03-30
>
> We thank the reviewer for the thorough and constructive review. We first address the questions (Q), followed by weaknesses (W).
> ## Q1: Ablations Across Problems and LLMs
> We run ablations on CVRP-ACO and across two LLM backbones, extending Table 3:
>
> **Table R1. Additional ablations on critic agents.**
> | | TSP-Constructive | CVRP-ACO | |
> |----|---|---|---|
> | Method | GPT-4o-mini (Gap↓) | GPT-4o-mini (Obj.↓) | GPT-5-nano low (Obj.) |
> | PathWise | 9.92% | 9.24 | 9.58 |
> | w/o Policy Critic | 13.31% | 9.32 | 10.85 |
> | w/o WM Critic | 12.07% | 9.27 | 9.90 |
> | w/o Policy Critic & WM Critic | 13.97% | 9.47 | 10.66 |
>
> Across both problems and LLMs: (1) removing the policy critic causes a larger drop than removing the world model (WM) critic, confirming the hierarchy in Table 3; (2) removing both generally causes the largest degradation.
> ## Q2: Entailment Graph vs. Flat History Log
> We conduct this ablation in detail; please see our response to `Reviewer xTWv` (Table R1), where we compare PathWise against two without-entailment-graph variants using the same multi-agent architecture: (1) Population + Historical Context, which preserves richer context in prompts but without graph structure, and (2) Population-Only, which excludes all historical context. Results show clear degradation from PathWise to both variants, confirming that the graph topology, not just richer context, drives improvement, with the entailment graph serving as the effective medium through which agents coordinate.
> ## Q3: Open-Source LLMs
> In addition to the closed-source results across 6 COPs in Tables 1-2, 7-9, we conducted experiments with the open-source model DeepSeek-V3.2:
>
> **Table R2. Gaps (%) of LLM-based AHD methods with DeepSeek-V3.2.**
>
> TSP-ACO:
> | Method | N=50 | N=100 |
> |---|---|---|
> | ReEvo | 3.53% | 8.59% |
> | HSEvo | 2.77% | 6.40% |
> | PathWise | 1.86% | 3.78% |
>
> KP-Constructive:
> | Method | N=50, W=12.5 | N=200, W=25 | N=500, W=25 |
> |---|---|---|---|
> | MCTS-AHD | 0.25% | 0.09% | 0.06% |
> | PathWise | 0.21% | 0.07% | 0.04% |
>
> PathWise also outperforms baselines with open-source LLMs on different COPs.
> ## Q4: API Stability
> We used fixed model endpoints (`gpt-4o-mini-2024-07-18`, `gpt-5-nano-2025-08-07`), both immutable. We account for sampling randomness by averaging 3 runs; please see our response to `Reviewer 1rPG` for statistical analysis with additional runs (all p < 0.05) across multiple COPs and LLMs.
> ## W1: Cost and Efficiency Analysis
> We provide detailed cost analysis in Appendix H, showing MRGI improvements at comparable cost. The three main cost dimensions in LLM-based AHD are: (1) **evaluation budget** $n_e$, fixed at 500 across all methods; (2) **training time**, dominated by heuristic execution; and (3) **monetary cost**, dominated by output tokens. API calls are lightweight web requests, not the bottleneck. For example, on Offline BPP-ACO with GPT-5-nano (low), PathWise and MCTS-AHD produce comparable tokens (801K vs. 800K) yet MCTS-AHD requires 2.46× longer training time (1.34 vs. 3.30 hrs).
> ## W2: Terminology and Originality
> Our use of "world model" follows [1], where an LLM predicts the next state given a current state and action via pretrained knowledge. In PathWise, the policy agent draws an action (parent selection + derivation rationale) from the current state using its internal knowledge, and the world model agent takes this action to create the update on the entailment graph (next state). This separation of planning (policy) and execution (world model) is supported by routed reflections from critic agents, where the policy critic reflects on evolutionary strategy and the world model critic reflects on heuristic generation quality. We acknowledge this distinction from classical learned-dynamics models and will clarify in our manuscript.
>
> Regarding originality, MCTS-AHD does not record derivation; its parent-child relationships are structural (UCT-based node selection via visit statistics). PathWise introduces a hybrid graph-based and population-based formulation, with the entailment graph adding semantic-level derivation history, routed reflections, and state pruning. Please see our response to `Reviewer upCx` (Q1) for further discussion on the MDP formulation and architectural distinction.
> ## W3: Gap to Specialized Solvers
> While this gap exists, LLM-based AHD offers two key advantages: (1) Generalizability: LKH-3 is a problem-specific solver, whereas PathWise discovers heuristics across different COPs and search frameworks. (2) Scaling: generated heuristics run orders of magnitude faster at scale:
>
> **Table R3. Runtime scaling of PathWise heuristics over LKH-3 on TSP (averaged over 50 instances, run 3 times).**
> | Test Set | Speedup |
> |---|---|
> | N=50 | 496× |
> | N=100 | 664× |
> | N=200 | 728× |
>
> This scaling advantage grows with problem size and, together with generalizability, is a key benefit of LLM-based AHD.
> ## References
> [1] S. Hao et al. "Reasoning with Language Model is Planning with World Model." arXiv:2305.14992

---

> > ### Author Rebuttal · Reviewer_rgzX · 2026-04-05
> >
> > Thanks, but I will keep my ratings.

---

> > > ### Author Response · Authors · 2026-04-06
> > >
> > > Thank you for acknowledging that our rebuttal fully addressed your concerns. We hope our clarifications and additional experiments will be reflected positively in your final assessment of the paper.

---

### Decision · Program_Chairs · 2026-04-30

**Decision:**

Accept (regular)

**Comment:**

This paper proposes a multi-agent framework for automated heuristic design, introducing an entailment graph to enable state-aware planning and structured memory. Reviewers agree that the problem is important and the empirical results are strong across multiple tasks and settings.

After rebuttal, most concerns were addressed, including additional ablations (especially isolating the entailment graph), experiments on open-source LLMs, and statistical validation. Three reviewers support acceptance (one accept, two weak accept). One reviewer maintains a weak reject mainly due to concerns about terminology (e.g., “world model”), rather than technical flaws.

Overall, the paper provides a solid and well-validated contribution. While the architectural novelty is moderate and terminology could be clearer, the entailment-graph formulation and consistent empirical gains make it a valuable addition to LLM-based heuristic design.